# Impact of individual and environmental factors on dietary or lifestyle interventions to prevent type 2 diabetes development: a systematic review

Dhanasekaran Bodhini[1], Robert W. Morton[2,3,4], Vanessa Santhakumar[5], Mariam Nakabuye[6], Hugo Pomares-Millan[7,8], Christoffer Clemmensen[6], Stephanie L. Fitzpatrick[9], Marta Guasch-Ferre[6,10], James S. Pankow[11], Mathias Ried-Larsen[12,13], Paul W. Franks[4,10,14,15], ADA/EASD PMDI*, Deirdre K. Tobias[5,10,202], Jordi Merino[6,16,17,202], Viswanathan Mohan[1,18,203] & Ruth J. F. Loos[6,19,203✉]

## Abstract

**Background** The variability in the effectiveness of type 2 diabetes (T2D) preventive interventions highlights the potential to identify the factors that determine treatment responses and those that would benefit the most from a given intervention. We conducted a systematic review to synthesize the evidence to support whether sociodemographic, clinical, behavioral, and molecular factors modify the efficacy of dietary or lifestyle interventions to prevent T2D.

**Methods** We searched MEDLINE, Embase, and Cochrane databases for studies reporting on the effect of a lifestyle, dietary pattern, or dietary supplement interventions on the incidence of T2D and reporting the results stratified by any effect modifier. We extracted relevant statistical findings and qualitatively synthesized the evidence for each modifier based on the direction of findings reported in available studies. We used the Diabetes Canada Clinical Practice Scale to assess the certainty of the evidence for a given effect modifier.

**Results** The 81 publications that met our criteria for inclusion are from 33 unique trials. The evidence is low to very low to attribute variability in intervention effectiveness to individual characteristics such as age, sex, BMI, race/ethnicity, socioeconomic status, baseline behavioral factors, or genetic predisposition.

**Conclusions** We report evidence, albeit low certainty, that those with poorer health status, particularly those with prediabetes at baseline, tend to benefit more from T2D prevention strategies compared to healthier counterparts. Our synthesis highlights the need for purposefully designed clinical trials to inform whether individual factors influence the success of T2D prevention strategies.

## Plain language summary

Clinical trials to prevent development of type 2 diabetes (T2D) that test dietary and lifestyle interventions have resulted in different results for different study participants. We hypothesized that the differing responses could be because of different personal, social and inherited factors. We searched different databases containing details of published research studies investigating this to look at the effect of these factors on prevention of the development of T2D. We found a small amount of evidence suggesting that those with poorer health, particularly those with a higher amount of sugar in their blood, tend to benefit more from T2D prevention strategies compared to healthier counterparts. Our results suggest that further clinical trials that are designed to examine the effect of personal and social factors on interventions for T2D prevention are needed to better determine the impact of these factors on the success of diet and lifestyle interventions for T2D.

---

A full list of author affiliations appears at the end of the paper.

Diabetes affects over 530 million people worldwide[1]. Around 90% of all diabetes is estimated to be type 2 diabetes (T2D), a non-autoimmune condition with marked pathophysiological heterogeneity[2]. In many cases, diet and physical activity interventions targeted at bodyweight reduction or preventing weight gain have demonstrated to delay progression[3–6], yet T2D remains a major cause of morbidity and mortality globally[7]. Chronic inadequate control of hyperglycemia causes downstream microvascular and macrovascular complications that drive the costly and debilitating T2D public health burden[7]. Coupled with its increasing incidence, public health and clinical efforts need to optimize effective upstream strategies for T2D prevention.

Landmark randomized intervention trials have demonstrated the effectiveness of intensive lifestyle interventions and glucose-lowering drug therapies for delaying the onset of T2D in patients at high risk[3–6]. However, T2D incidence has only escalated in the decades since, despite the success of early clinical trials. Thus, implementation strategies for diabetes prevention in the real-world setting involving more practical ways of identifying high-risk individuals and precision prevention research may contribute to understanding this gap[8].

Precision prevention of T2D serves to minimize an individual's T2D risk factor profile and maximize the effectiveness of new or established strategies for disease prevention through targeting biological interactions and/or removing barriers to access and adherence to lifestyle modification[9]. For example, precision prevention approaches might use clinical (e.g., age, sex, body mass index [BMI]), social (e.g., education attainment, socioeconomic status), or molecular (e.g., genetic, 'omic' traits) characteristics to inform strategies likely to elicit the most effective or sustainable response for an individual, resulting in tailored prevention strategies[9–11].

The purpose of this systematic review is to critically appraise the accumulated experimental evidence underpinning the feasibility and effectiveness of the clinical translation of precision prevention of T2D. The scope of our investigation included studies reporting the effect modification of lifestyle and dietary interventions for T2D prevention by any of the following individual-level factors, including sociodemographics, clinical risk factors, behavior, or molecular traits. This work was undertaken as part of a series of systematic reviews conducted by the ADA/EASD Precision Medicine in Diabetes Initiative[12], an international collaboration of global leaders in precision diabetes medicine[13].

Through this systematic review, we found low certainty evidence that those with poorer health status, particularly those with prediabetes at baseline, tend to benefit more from T2D prevention strategies compared to healthier counterparts. Clinical trials specifically designed to inform whether individual factors influence the success of T2D prevention strategies are needed in the future.

## Methods
The systematic review protocol was pre-registered on the International Prospective Register of Systematic Reviews (PROSPERO; CRD42021267686).

**Data sources and search**. Our search included MEDLINE, Embase, and Cochrane Central Register of Controlled Trials databases for studies reporting on the efficacy of lifestyle or behavioral interventions with T2D incidence, published from 1/1/2000 to 7/15/2021. Lifestyle interventions were defined as interventions ranging from interventions on single behavioral factors including diet, physical activity, smoking, and body weight loss, to multi-component modification programs focused on different behavioral components. An experienced librarian developed a search strategy (Supplementary Note 1), which included combinations of keywords related to lifestyle intervention for preventing T2D (diet, lifestyle, physical activity, body weight), study design, and health outcome, and was limited to the English language. We also scanned the references of included manuscripts and the reference list of systematic reviews published within the past 2 years to identify additional relevant studies.

**Study selection**. We included studies reporting the effect of a lifestyle, dietary pattern, or dietary supplement interventions vs. other active comparators or control on the incidence of T2D and reporting the results stratified by any eligible factor. Lifestyle interventions included either single-component (exercise, smoking, education through text messaging to the mobile phone, etc) or multi-component modification programs involving weight loss through diet or supplementation, physical activity, awareness education etc. Eligible stratification factors, or effect modifiers, included individual-level sociodemographic (i.e., race/ethnicity, socioeconomic status/education, location, age, sex), clinical factors (i.e., BMI, dysglycemia, presence of comorbidities), behavioral (i.e., baseline diet, physical activity) or molecular traits (i.e., genetics, metabolites). We did not review population-level exposures such as built environment, pollution, or climate. Off-label pharmaceutical interventions and bariatric surgery were beyond the scope of the review. We limited inclusion to studies in adults aged >18 years and enrolling at least 100. We included non-randomized and randomized clinical studies delivering an eligible intervention, comparing against another active intervention, usual care, placebo control, or non-control group. The majority of studies ($N = 76$ or 94%) included in this review are RCTs to examine the effect on the intervention on T2D incidence. However, as our focus is on the modification of the intervention effect by sociodemographic, clinical, behavioral and molecular factors, none of these trials can be considered randomized for the purpose of this review, as the randomization block is not conserved. Studies exclusively among individuals with a current or history of gestational diabetes were excluded because they overlapped in scope with another PMDI consortium review.

**Screening, data extraction, and quality assessment**. We used the Covidence online systematic review platform[14] for literature screening, data extraction, and consensus. Screening consisted of two stages: (1) title and abstract and (2) full text. At each screening stage, two independent reviewers determined the eligibility of the citation, and in the case of disagreement, a third reviewer resolved the discrepancy. Among the full papers accepted for inclusion in the review, two independent reviewers extracted detailed information on the study design, participant characteristics, interventions, comparators, effect modifiers, follow-up for T2D, and analytic approach. We extracted findings related to the effect modification of treatment vs. comparator on T2D risk, including strata-specific treatment groups' T2D cases and incidence rates, or strata-specific treatment-comparator incidence rate ratios, relative risks, risk differences, etc., including measures of variance. We also extracted data on different available measurements for the interaction of the effect modifier with the intervention effect on T2D, including interaction term estimates, interaction term p-value, stratified estimates, heterogeneity test and noted any text referring to tests performed with "data not shown". We developed and piloted the data extraction template (Supplementary Table 1), and discrepancies were ruled on by a third reviewer. The relevant statistical results extracted for each effect modifier has been provided as Supplementary Data 1.

We evaluated the studies' risk of bias using a modified JBI Critical Appraisal Checklist for randomized controlled trials[15], performed by two independent reviewers and disagreements resolved by a third reviewer. We modified the 13-item checklist to 9 questions tailored to evaluating the quality of the study design but with consideration for our primary interest in stratified results rather than the total intervention effect for T2D risk. These 9 questions were mainly based on randomization, interventions, treatment, and assessor blindness to outcome assessment. Our evaluation corresponded to color coding in a heat map organized by intervention type and effect modifier (Supplementary Fig. 1).

**Synthesis of results**. We collated the literature according to intervention type as lifestyle intervention programs (single or multi-component), dietary pattern interventions (involving modifications in diet only), or supplement intervention and effect modifier analyzed (e.g., sex, age strata) to synthesize results. We determined that a meta-analysis was not feasible among the studies included in our review due to paucity and marked differences in the nature of the study populations, interventions and comparators, study designs, and effect modifiers analyzed. We qualitatively evaluated the direction and magnitude of results and statistical tests among each prevention strategy for each effect modifier. We weighed these qualitative and quantitative results against their risk of bias. We qualitatively synthesized the evidence for each modifier based on the direction of findings reported in available studies. We used the Diabetes Canada Clinical Practice Scale to assess the certainty of the evidence for a given effect modifier[16]. A level of evidence was assigned following the approach and criteria described in Supplementary Table 2. For example, higher levels were assigned if the study was a systematic overview or meta-analysis of high-quality RCTs or an appropriately designed RCT with adequate power to answer the question posed by the investigators. Then, each recommendation was assigned a grade from A to D. Two reviewers independently assessed the certainty of the evidence and resolved disagreements through consensus discussion.

**Reporting summary**. Further information on research design is available in the Nature Portfolio Reporting Summary linked to this article.

## Results
The results of our systematic literature search are presented in the Fig. 1 PRISMA flow diagram. Of the 10,880 citations identified through database searches and other sources, 1047 abstracts were retrieved for full-text review. From these, 81 publications met our inclusion criteria, and data were extracted.

**Study characteristics**. The 81 publications included in our review represented 33 unique intervention studies (Table 1 and Supplementary Table 3). Twenty-eight studies were randomized clinical trials (RCTs), three were nonrandomized parallel group trials, and two were single-arm clinical interventions. Fourteen intervention studies took place in Asia, 11 in Europe, seven in North America, and one was a multicenter study that took place in Asia and Europe. Intervention enrollment sample sizes ranged from 302 to 48,835 participants (Table 1). Twenty-two studies included individuals at high risk for T2D, two studies at increased cardiovascular risk, and other studies included the general population or other specific groups. The active intervention times ranged from one lifestyle counseling visit to active interventions lasting up to 10 years (Supplementary Fig. 2).

Twenty-four of the included studies assessed the effect of a multi-component lifestyle intervention program focused on changes in diet, physical activity, smoking, or body weight loss. Four studies implemented a dietary intervention, and five administered supplements. Across multi-component lifestyle intervention studies, the comparator consisted of a less intensive lifestyle program consisting of usual care or general lifestyle advice administered at baseline. Active comparator groups for

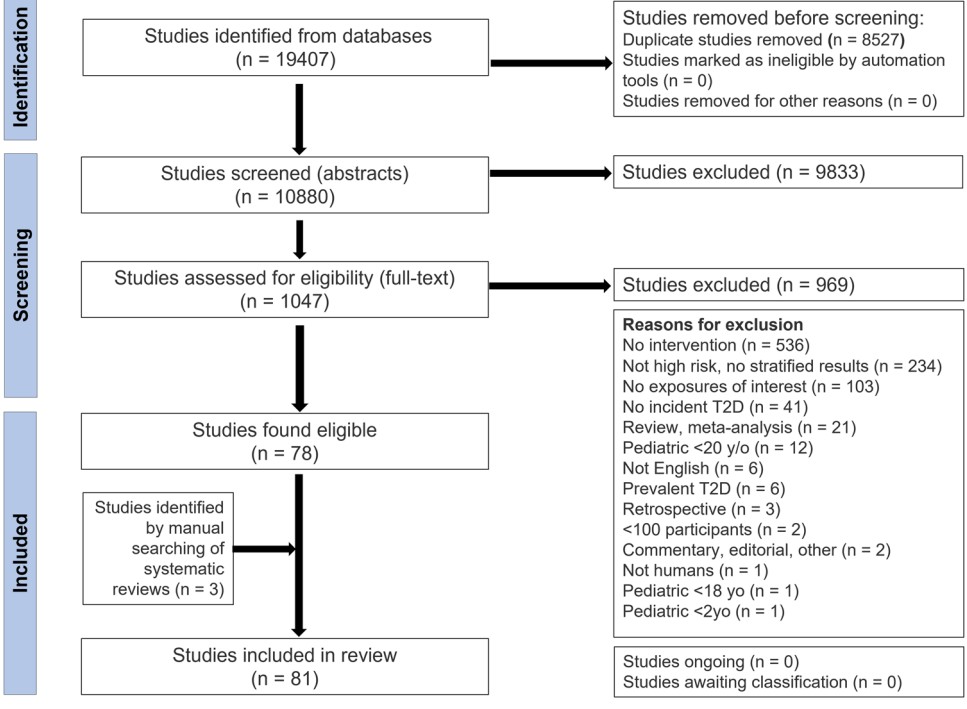

**Fig. 1 PRISMA flow diagram.** Stepwise screening stages adapted for selecting the studies of interest using Covidence software. Screening at all stages was done by two independent reviewers, and a third reviewer resolved conflicts.

**Table 1 Description of study population and study design of the included trials grouped according to the type of intervention.**

| Trial/Study name | Country | Total enrolled; inclusion criteria | Baseline enrollment years | Intervention design | Active intervention duration | Intervention(s) | Comparator/control intervention | Main trial info PMIDs | Included studies (PMIDs) |
|---|---|---|---|---|---|---|---|---|---|
| **Lifestyle interventions** | | | | | | | | | |
| Chae et al.[27] | South Korea | N = 7233; General population | 2007/11 | Non-randomized, parallel arm | 6 months | Physical activity program | Usual care | 22688549[27] | 22688549[27] |
| Da Qing IGT and Diabetes Study | China | N = 577; Prediabetes | 1986 | Cluster-randomized trial | 6 years | (i) Diet: Low-calorie, low-fat (25–30% kcal) healthy pattern; (ii) Increase exercise; (iii) Diet + Exercise | Provided with diabetes education materials at baseline | 9096977[5] | 24731674[28], 34212465[17], 12413779[29] |
| Diabetes Community Lifestyle Improvement Program (D-CLIP) | India | N = 578; Prediabetes | 2009/12 | Randomized, parallel arm | 3 years | Lifestyle diabetes prevention program + Metformin | Lifestyle diabetes prevention program at baseline | 27504014[18] | 27504014[18] |
| Diabetes in Europe—Prevention using Lifestyle, Physical Activity and Nutritional - Catalonia (DE-PLAN-CAT)[a] | Spain | N = 544; Prediabetes | 2006 | Non-randomized, parallel arm | 4 years | Lifestyle weight loss and diabetes prevention program | Provided with diabetes education materials at baseline | 22322921[30] | 22322921[30] |
| Diabetes Prevention Program (DPP)[a] | US | N = 3234; Prediabetes | 1996/99 | Randomized, parallel arm | Mean 2.8 years | (i) Lifestyle weight loss and diabetes prevention program; (ii) Metformin | Usual care + placebo | 11832527[3] | 26024851[31], 33444158[32], 33176293[33], 28453780[34], 19640960[35], 23860722[36], 16855264[37], 19017751[38], 18060660[39], 11832527[3], 23512951[40], 17077202[41], 19878986[42], 33394545[43], 29021207[44], 21378175[45], 20682687[46], 17363740[47], 25697494[48], 25277388[49], 23451166[51] |
| EDIPS-Newcastle | UK | N = 102; Prediabetes | | Randomized, parallel arm | 5 years | Lifestyle diabetes prevention program | Usual care | 19758428[50] | |
| Finnish Diabetes Prevention Study (DPS)[a] | Finland | N = 522; Prediabetes | 1993/98 | Randomized, parallel arm | Mean 3.2 years | Lifestyle and weight loss diabetes prevention program | Provided with diabetes education materials at baseline | 11333990[4] | 16759313[52], 17277585[53], 16873669[54], 18249219[55], 19651919[56], 17636114[57], 11333990[4], 20980412[58], 21451749[59], 15127203[60], 18252900[61], 15616024[62], 15309292[63], 15983230[64], 18091023[65], 15126514[66], 17437080[67] |
| Indian Diabetes Prevention Program 2013 (IDPP-2013) | India | N = 537; Men with prediabetes | 2009 | Randomized, parallel arm | 24 months | SMS-delivered lifestyle diabetes prevention education | Provided with diabetes education materials at baseline | 24622367[68] | 26773871[69], 16391903[6] |

**Table 1 (continued)**

| Trial/Study name | Study population | | Study design | | | Intervention(s) | Comparator/control intervention | Main trial info PMIDs | Included studies (PMIDs) |
|---|---|---|---|---|---|---|---|---|---|
| | Country | Total enrolled; inclusion criteria | Baseline enrollment years | Intervention design | Active intervention duration | | | | |
| Indian Diabetes Prevention Programme (IDPP-1) | India | N = 531; Prediabetes | 2001/02 | Randomized, parallel arm | 3 years | (i) Lifestyle diabetes prevention program; (ii) Metformin; (iii) Lifestyle + Metformin | Usual care | 16391903[6] | 16391903[6], 20519663[70], 26773871[69] |
| Indian Diabetes Prevention Programme (IDPP-2) | India | N = 407; Prediabetes | 2003/05 | Randomized, parallel arm | 3 years | Lifestyle diabetes prevention program + Pioglitazone | Lifestyle diabetes prevention program + Placebo | 19277602[71] | 20519663[70] |
| Japan Diabetes Prevention Program (Japan DPP)[a] | Japan | N = 304; Prediabetes | 1999/02 | Randomized, parallel arm | 3 years | Lifestyle weight loss and diabetes prevention program | Provided with diabetes education materials at baseline | 25452854[72] | 25452854[72], 21235825[20] |
| Kerala Diabetes Prevention Program (K-DPP)[a] | India | N = 1007; Prediabetes, rural | 2013 | Cluster-randomized trial | 12 months | Peer-led lifestyle diabetes prevention program | Provided with diabetes education materials at baseline | 24180316[73] | 29872436[74] |
| Kosaka et al.[75] | Japan | N = 458; Men with prediabetes | 1990/92 | Randomized, parallel arm | 4 years | Lifestyle weight loss and diabetes prevention program | Lifestyle weight loss and diabetes prevention information only | 15649575[75] | 15649575[75] |
| Let's Prevent Diabetes | UK | N = 880; Prediabetes | 2009/11 | Cluster-randomized trial | 36 months | Lifestyle diabetes prevention program | Provided with diabetes education materials at baseline | 22607160[76] | 26740346[77] |
| Multiple Risk Factor Intervention Trial (MRFIT) | US | N = 12,866; Men with high cardiovascular risk | 1973/76 | Randomized, parallel arm | 6 years | Lifestyle modifications for heart disease prevention | Usual care | 15738450[78] | 15738450[78] |
| Nanditha et al.[79] | India, UK | N = 2062; Prediabetes | 2012/17 | Randomized, parallel arm | 24 months | SMS-delivered lifestyle diabetes prevention education | Provided with diabetes education materials at baseline | 31919539[79] | 31919539[79] |
| National Program for the Prevention of Type 2 Diabetes (FIN-D2D)[a] | Finland | N = 2798; Prediabetes | 2004/07 | Population-wide intervention | Mean 14 months | Lifestyle weight loss and diabetes prevention program | - | 20664020[80] | 22983785[81], 33771515[82], 21781153[83], 20664020[80], 21262677[84], 34177805[85] |
| Niyantrita Madhumeha Bharata Abhiyaan (NMB-Trial) | India | N = 4450; Prediabetes | 2017 | Cluster-randomized trial | 3 months | Yoga-based lifestyle diabetes prevention program | Presentation on lifestyle for diabetes prevention at baseline | 34177805[85] | 34177805[85] |
| Norfolk Diabetes Prevention Study (NDPS) | UK | N = 1028; Prediabetes | 2011/18 | Randomized, parallel arm | 12–46 months | (i) Lifestyle diabetes prevention program; (ii) Lifestyle diabetes prevention program with peer support | Provided with diabetes education materials at baseline | 33136119[86] | 33136119[86] |
| Prevention of Diabetes in Euskadi (PreDE)[a] | Spain | N = 1088; Prediabetes | 2011/13 | Cluster-randomized trial | 24 months | Lifestyle weight loss and diabetes prevention program | Usual care | 29476888[87] | 29476888[87] |
| Tehran Lipid and Glucose Study (TLGS) | Iran | N = 10,368; General population | 1999/01 | Non-randomized, cluster intervention | Mean 3.6 years | Lifestyle program for chronic disease prevention | Usual care | 20494239[88] | 25029368[89], 20494239[88] |
| Thai Diabetes Prevention Program (Thai DPP) | Thailand | N = 1903; Prediabetes | 2013 | Cluster-randomized trial | 24 months | Lifestyle diabetes prevention program | Provided with diabetes education materials at baseline | 31079517[19] | 31079517[19] |
| Västerbotten Intervention Programme (VIP)[a] | Sweden | N = 113, 203; General population | 1987-present | Population-wide intervention | Ongoing | Lifestyle CVD and diabetes prevention program | - | 20339479[90] | 25532678[91] |
| Zensharen Study for Prevention of Lifestyle Diseases[a] | Japan | N = 641; Prediabetes | 2004/06 | Randomized, parallel arm | 36 months | Lifestyle weight loss program + Frequent engagement | Lifestyle weight loss program + Minimal engagement | 21824948[92] | 21824948[92] |
| **Dietary pattern interventions** | | | | | | | | | |
| CORonary Diet Intervention with Olive oil and cardiovascular PREVention study (CORDIOPREV) | Spain | N = 1002; Prevalent heart disease | 2009/12 | Randomized, parallel arm | Median 7 years | Mediterranean dietary pattern | AHA low-fat pattern (<30% kcal) | 27297848[93] | 32723508[94] |

**Table 1 (continued)**

| Trial/Study name | Study population | | Study design | | | Intervention(s) | Comparator/control intervention | Main trial info PMIDs | Included studies (PMDIs) |
|---|---|---|---|---|---|---|---|---|---|
| | Country | Total enrolled; inclusion criteria | Baseline enrollment years | Intervention design | Active intervention duration | | | | |
| Primary Prevention of Cardiovascular Disease with a Mediterranean Diet Supplemented with Extra-Virgin Olive Oil or Nuts (PREDIMED) | Spain | N = 7447; High cardiovascular risk | 2003/09 | Randomized, parallel arm | 4.8 years | (i) Mediterranean pattern + extra-virgin olive oil; (ii) Mediterranean pattern + mixed nuts | Low-fat pattern | 29897866[95] | 29663011[96], 26739996[97], 23034962[98], 31371719[99], 24573661[100], 20929998[101] |
| Shahbazi et al. [102] | Iran | N = 336; Prediabetes | 2012 | Randomized, parallel arm | 2 years | (i) High-fat diet from olive oil (45% kcal); (ii) Normal fat diet (30% kcal) | Standard low-fat diet (<30% kcal) | DOI 10.1007/s13410-017-0548-3 | DOI 10.1007/s13410-017-0548-3[102] |
| Women's Health Initiative Dietary Modification Trial (WHI-DM) | US | N = 48,835; Healthy postmenopausal women | 1993/98 | Randomized, parallel arm | Mean 8.1 years | Low-fat (20% kcal) healthy pattern | Provided with healthy diet materials at baseline | 18663162[103] | 29282203[104] |
| **Dietary supplement interventions** | | | | | | | | | |
| Alpha-Tocopherol, Beta-Carotene Lung Cancer Prevention Study (ATBC) | Finland | N = 29,133; Men, smokers | 1985/88 | Randomized, parallel arm | Median 6.1 years | 2 × 2 factorial: (i) alpha-tocopherol (50 mg/day), (ii) beta-carotene (20 mg/day) | Placebo | 8205268[105] | 17994292[106] |
| Vitamin D and Type 2 Diabetes Trial (D2d) | US | N = 2423; Prediabetes | 2013/17 | Randomized, parallel arm | Median 2.5 years | Vitamin D supplementation (4000 IU/day) | Placebo | 31173679[107] | 31173679[107] |
| Women's Antioxidant and Folic Acid Cardiovascular Study (WAFACS) | US | N = 5442; Women with cardiovascular disease | 1998 | Randomized, parallel arm | Median 7.3 years | Folic acid (2.5 mg/day), vitamin B6 (50 mg/day), and vitamin B12 (1mg/day) combined supplementation | Placebo | 19491213[108] | 19491213[108] |
| Women's Antioxidant Cardiovascular Study (WACS) | US | N = 8171; Women with cardiovascular disease | 1995/96 | Randomized, parallel arm | Median 9.2 years | 2 × 2 × 2 factorial: (i) vitamin C (500 mg/day), (ii) vitamin E (600 IU/day), (iii) beta-carotene (50 mg/eod) supplementation | Placebo | 19491386[109] | 19491386[109] |
| Women's Health Study (WHS) | US | N = 39,876; Healthy women | 1992/95 | Randomized, parallel arm | 10.1 years | 2 × 2 Factorial, every other day: Aspirin (100 mg); (ii) Vitamin E supplementation (600 IU) | Placebo | 15998891[110] | 17003353[111] |

aTrials which aimed at weight loss and prevention of T2D.

**Table 2 Efficacy of T2D preventive interventions according to sociodemographic effect modifiers.**

| Modifier | T2D preventive strategies | | | | | | | | |
| | Lifestyle intervention | | | Dietary pattern intervention | | | Dietary supplements intervention | | |
| | Number of studies | Effect modification[a] | Certainty of evidence[b] | Number of studies | Effect modification[a] | Certainty of evidence[b] | Number of studies | Effect modification[a] | Certainty of evidence[b] |
|---|---|---|---|---|---|---|---|---|---|
| Age | 12 | Yes: 7 studies No: 5 studies | Grade D | 3 | No: 3 studies | Grade D | 4 | Yes: 1 study No: 3 studies | Grade D |
| Sex | 16 | Yes: 1 study No: 15 studies | Grade D | 2 | No: 2 studies | Grade D | 1 | Yes: 1 study | Grade D |
| Race/ethnicity | 3 | No: 3 studies | Grade D | 1 | No: 1 study | Grade D | 1 | No: 1 study | Grade D |
| Socioeconomic status/ Education | 4 | Yes: 1 study No: 3 studies | Grade D | – | – | – | – | – | – |
| Location | 2 | No: 2 studies | Grade D | – | – | – | 1 | No: 1 study | – |

Overview of the included studies investigating whether sociodemographic factors modify the response to T2D preventive intervention strategies.
[a]Yes/No corresponds to significant/nonsignificant effect modification, as reported in the study.
[b]Certainty of evidence denotes consistency, Grading based on Diabetes Canada scale A to D.

dietary intervention studies focused on high-fat diets consisted of a low-fat intervention. The active comparator for supplement studies consisted of a placebo intervention. T2D was diagnosed in person with an oral glucose tolerance test (OGTT) in 27 studies, whereas in 6 studies, T2D was ascertained via self-report or through linkage with a healthcare registry database. The primary endpoint was T2D incidence in 21 studies or a composite cardiovascular event in six studies (Table 1 and Supplementary Table 3).

All except seven studies of a multi-component lifestyle intervention program showed evidence that a lifestyle intervention reduces the risk of T2D, with estimated relative risk reduction ranging from 60 to 23% (Supplementary Table 3). Available evidence also suggests that a high-fat diet (Mediterranean pattern diet with extra-virgin olive oil/ mixed nuts or high-fat diet from olive oil), reduces the relative risk of T2D when compared to a diet with a lower amount of fat. Evidence from studies using supplements showed a null effect on T2D risk reduction.

Our certainty of evidence assessment determined that the primary study design and approach was generally low, particularly for the RCTs, owing to randomization methods and uniform outcome assessment (Supplementary Fig. 1). However, common concerns for bias were due to non-blinding of participants, deliverers, and outcomes assessors to treatment assignment. Nonrandomized interventions and RCTs having additional concerns for study design did have ratings of high risk of bias.

**Sociodemographic and clinical factors**. Some clinical trials, such as the Diabetes Prevention Program (DPP), the Finnish Diabetes Prevention Study (DPS), or the PREDIMED study, were highly represented, with 20, 16, and 6 different publications from each study, respectively. Certainty of evidence to indicate different effects for sociodemographic and clinical characteristics such as age, sex, race/ethnicity, socioeconomic status or geographic location in response to lifestyle intervention was low. Study-specific numeric estimates for the effect modification are provided in the extended data file. Evidence from studies investigating sociodemographic interaction effects in dietary modification or supplementation trials showed no significant heterogeneity in response to intervention according to these characteristics (Table 2 and Fig. 2).

Fourteen studies investigated whether BMI modified the efficacy of multi-component lifestyle interventions. Nine of these studies showed that BMI is not associated with different responses to a lifestyle program, but five studies showed suggestive evidence that individuals with low BMI could benefit most from a lifestyle intervention. Four of these five studies presenting evidence of the differential effect of a lifestyle intervention according to BMI were conducted in Asia (Table 3). No appreciable evidence for interactions with BMI was observed in studies that implemented a dietary or supplement intervention (Table 3). Eighteen studies tested the efficacy of an intensive lifestyle intervention for preventing T2D stratified based on baseline glucose levels, impaired glucose tolerance, or prediabetes status. Evidence presented in eight of these studies indicated statistically different effects based on baseline dysglycemia, but other studies did not find evidence of effect modifications. Three studies investigated family history of T2D as a potential lifestyle intervention effect modifier, and only one provided suggestive evidence of heterogenous treatment responses. Studies stratified by baseline cardiometabolic risk factors reported that individuals with poorer health status, particularly those with dyslipidemia and metabolic syndrome, tend to benefit more from dietary or supplement interventions than healthier individuals (Table 3).

**Behavioral factors**. Several secondary studies have assessed whether baseline lifestyle factors (i.e., overall dietary quality, alcohol intake, physical activity, and/or smoking) influence the efficacy of T2D prevention interventions. Evidence presented in studies investigating the effect of a lifestyle intervention according to baseline smoking status and physical activity indicates statistically different effects, suggesting that smokers and those with lower physical activity levels benefited less from a lifestyle program (Table 4). Available studies reported no interactions of baseline smoking status and physical activity levels with dietary or supplement interventions on the risk of T2D. Among the four studies that focused on alcohol intake, only one found that the lifestyle intervention was more effective in individuals who drink alcohol frequently than in those who rarely drink. Six studies tested whether baseline diet modified the association between supplements and the risk of T2D and found no evidence of significant interactions (Table 4).

**Molecular factors**. The extent to which genetic predisposition modifies the efficacy of interventions to prevent T2D was reported in 22 publications. Most of them were based on data

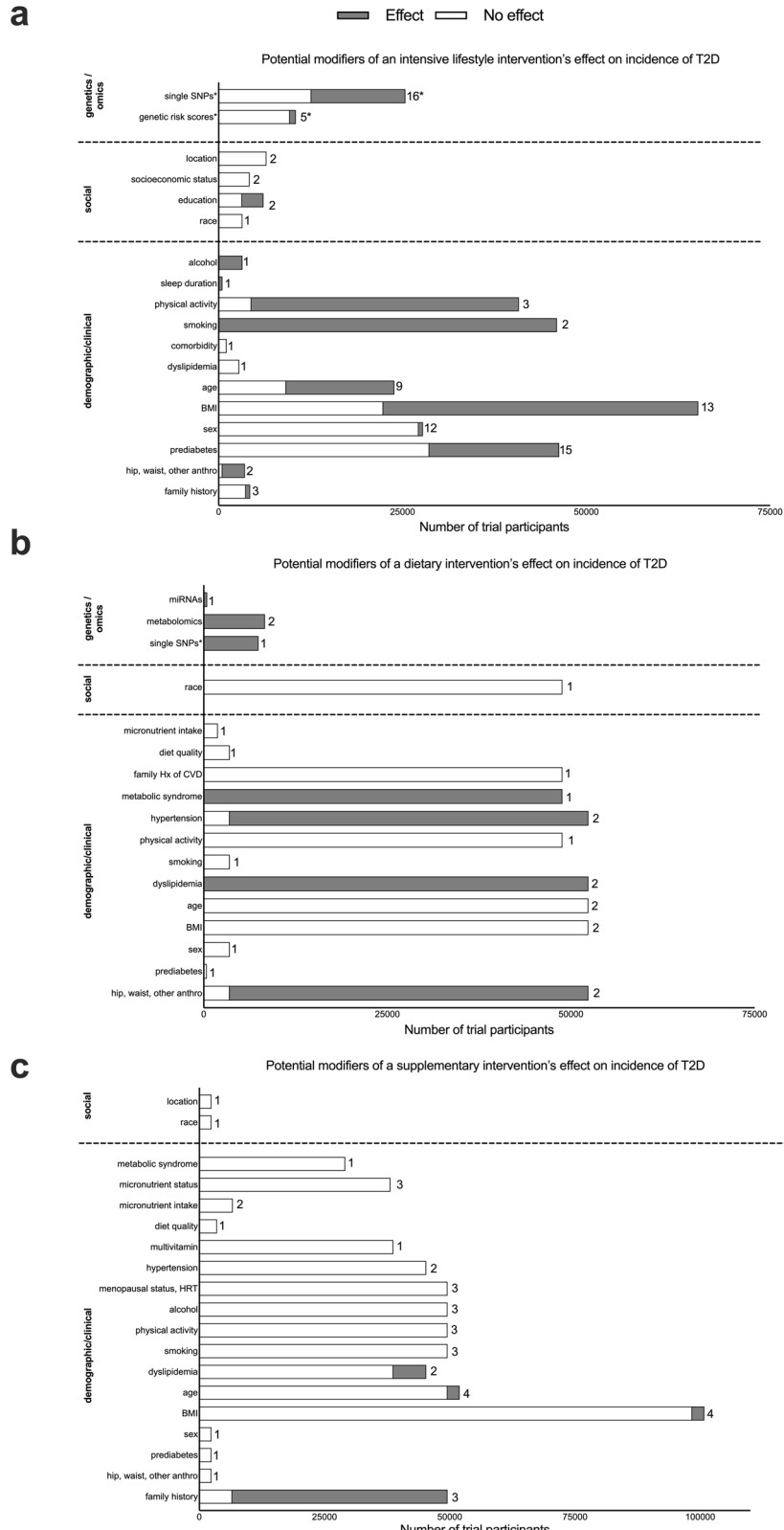

**a** Potential modifiers of an intensive lifestyle intervention's effect on incidence of T2D

**b** Potential modifiers of a dietary intervention's effect on incidence of T2D

**c** Potential modifiers of a supplementary intervention's effect on incidence of T2D

from the DPP and the DPS. Genetic predisposition was defined based on single genetic variants in 17 studies or genetic risk scores in five. While many of the T2D-associated loci identified in the earlier GWAS studies have been examined for their potential roles as effect modifiers, some reported evidence that individuals with specific genotypes could benefit the most from a lifestyle intervention, but these studies rarely corrected for the number of

performed tests. Of the five studies that reported on the role of polygenic scores for T2D, only one study showed that lifestyle intervention was more effective among individuals with a high genetic risk.

Besides genetics, other molecular markers such as plasma branched-chain amino acids and miRNAs have been studied. The evidence that these molecular features modify the efficacy of

**Fig. 2 Potential effect modifiers of lifestyle, diet, and diet supplements intervention on the incidence of T2D.** General overview of potential effect modifiers of lifestyle (**a**), dietary (**b**), and supplement (**c**) interventions on the incidence of type 2 diabetes. The *Y* axes indicate potential effect modifiers, and the *X* axes illustrate the total number of trial participants included in the studies investigating each modifier. The proportion of gray or white in each bar indicates the number of trial participants included in the studies where there was (gray) or was not (white) an effect by the effect modifier. Caution is warranted because whether an effect modifier did (or did not) have an effect is based on statistical significance from the publication's summary statistics. It is improbable that the effect modifier strictly did (or did not) have an effect on every participant included in that publication. The number of trials and trial participants are plotted because some trials (e.g., DPP) had multiple studies published using the same participants, so that the participant number would be heavily skewed. There was no instance where the same trial had multiple published studies evaluating the same effect modifier showing different results (e.g., there was no difference between sexes on the PREDIMED trial's effect on T2D incidence in their primary vs. subgroup studies/publications). The number at the end of each bar represents the number of trials for each potential effect modifier. *indicates an exception for genetics because the effect modifiers (SNPs or GRS) were all uniquely distinct but are presented together under the categories of "SNP" or "GRS" here.

**Table 3 Efficacy of T2D preventive interventions according to clinical effect modifiers.**

| Modifier | T2D preventive strategies | | | | | | | | |
| | Lifestyle intervention | | | Dietary pattern intervention | | | Dietary supplements intervention | | |
| | Number of studies | Effect modification[a] | Certainty of evidence[b] | Number of studies | Effect modification[a] | Certainty of evidence[b] | Number of studies | Effect modification[a] | Certainty of evidence[b] |
|---|---|---|---|---|---|---|---|---|---|
| BMI | 14 | Yes: 5 studies No: 9 studies | Grade D | 3 | No: 3 studies | Grade D | 4 | Yes: 1 study No: 3 studies | Grade D |
| Prediabetes | 18 | Yes: 8 studies No: 10 studies | Grade D | 1 | No: 1 study | Grade D | 1 | No: 1 study | Grade D |
| Family history | 3 | Yes: 1 study No: 2 studies | Grade D | – | – | – | 3 | Yes: 2 studies No: 1 study | Grade D |
| Dyslipidemia/ medications | 1 | No: 1 study | Grade D | 2 | Yes: 2 studies | Grade D | 2 | Yes: 1 study No: 1 study | Grade D |
| Hypertension | – | – | – | 2 | Yes: 1 study No: 1 study | Grade D | 2 | No: 2 studies | Grade D |
| Metabolic syndrome | – | – | – | 1 | Yes: 1 study | Grade D | 1 | No: 1 study | Grade D |
| Menopausal status, HRT use | – | – | – | – | – | – | 3 | No: 3 studies | Grade D |

Overview of the included studies investigating whether clinical factors modify the response to T2D preventive intervention strategies.
[a]Yes/No corresponds to significant/nonsignificant effect modification, as reported in the study.
[b]Certainty of evidence denotes consistency, Grading based on Diabetes Canada scale A to D.

**Table 4 Efficacy of T2D preventive interventions according to behavioral effect modifiers.**

| Modifier | T2D preventive strategies | | | | | | | | |
| | Lifestyle intervention | | | Dietary pattern intervention | | | Dietary supplements intervention | | |
| | Number of studies | Effect modification[a] | Certainty of evidence[b] | Number of studies | Effect modification[a] | Certainty of evidence[b] | Number of studies | Effect modification[a] | Certainty of evidence[b] |
|---|---|---|---|---|---|---|---|---|---|
| Smoking | 2 | Yes: 2 studies | Grade D | 1 | No: 1 study | Grade D | 3 | No: 3 studies | Grade D |
| Physical activity | 3 | Yes: 2 studies No: 1 study | Grade D | 1 | No: 1 study | Grade D | 3 | No: 3 studies | Grade D |
| Alcohol intake | 1 | Yes: 1 study | Grade D | – | – | – | 3 | No: 3 studies | Grade D |
| Diet and supplements | – | – | – | 2 | No: 2 studies | Grade D | 6 | No: 6 studies | Grade D |

Overview of the included studies investigating whether behavioral factors at baseline modify the response to T2D preventive intervention strategies.
[a]Yes/No corresponds to significant/nonsignificant effect modification, as reported in the study.
[b]Certainty of evidence denotes consistency, Grading based on Diabetes Canada scale A to D.

dietary interventions in the prevention of T2D has only low to very-low certainty (Table 5 and Fig. 2).

**Grading of evidence certainty**. Although our systematic review included intervention studies, most RCTs with low risk of bias, we evaluated certainty through our hypothesis of identifying valid effect modifiers to inform precision prevention. None of the studies included a priori consideration of intervention interactions with individual-level characteristics or risk factors in their study design, which were largely conducted as post hoc analyses. As a result, statistical power was often limited. Further, most did not adjust for individual-level risk factors, undermining the validity of interpreting effect modifiers' role independent of other traits. These considerations were factored into the major downgrading of the evidence (Tables 2–5).

**Table 5 Efficacy of T2D preventive interventions according to molecular effect modifiers.**

| | T2D preventive strategies | | | | | |
| --- | --- | --- | --- | --- | --- | --- |
| | Lifestyle intervention | | | Dietary pattern intervention | | |
| Modifier | Number of studies | Effect modification[a] | Certainty of evidence[b] | Number of Studies | Effect modification[a] | Certainty of evidence[b] |
| T2D single SNPs | 17 | Yes: 9 studies No: 7 studies Not reported: 1 study | Grade D | 1 | Yes: 1 study | Grade D |
| Diabetes polygenic score | 5 | Yes: 1 study No: 4 studies | Grade D | – | – | – |
| Metabolites/miRNA | – | – | – | 3 | Yes: 3 studies | Grade D |

Overview of the included studies investigating whether genetic and molecular factors at baseline modify the response to T2D preventive intervention strategies.
[a]Yes/No corresponds to significant/nonsignificant effect modification, as reported in the study.
[b]Certainty of evidence denotes consistency, Grading based on Diabetes Canada scale A to D.

## Discussion

We performed a comprehensive systematic review to identify individual-level sociodemographic, clinical, behavioral, or molecular factors that could modify the efficacy of T2D prevention strategies. Overall, we find low to very low certainty of evidence that traits such as age, sex, BMI, race/ethnicity, socioeconomic status, baseline lifestyle factors, or genetics consistently and validly modify the effectiveness of lifestyle and behavioral interventions. Individuals with prediabetes at baseline benefit slightly more from prevention interventions than those without prediabetes, but the certainty of the evidence was low. This can be explained by relative and absolute risk differences among people with/without prediabetes. However, whether the modest benefit reported in these studies was due to poor health status or other correlated risk factors cannot be ascertained based on the available evidence.

Large randomized clinical trials have consistently demonstrated that a healthy lifestyle or dietary interventions can prevent or delay T2D[3,4,6,17]. However, there is large inter-individual variability in response to these preventive interventions, in which some people seem to greatly benefit from T2D preventive interventions. Precision prevention aims to identify participant characteristics that determine this variability in response to ultimately tailor preventive strategies to subgroups of individuals that are likely to benefit the most. So far, no studies exist that were prospectively designed to determine interactions by a baseline trait or factor with an intervention to prevent T2D. We evaluated the evidence base and identified several stratified post hoc analyses of existing prevention intervention trials. In post hoc analyses, the participant population is stratified by a potential effect modifier, and the efficacy of the intervention is tested within each stratum and compared across the strata, which reduces statistical power and increases type 2 error.

Furthermore, precision prevention strategies may be optimized by incorporating several individual-level factors into decision-making, whereas the current literature predominantly evaluates one stratified trait at a time. For example, correlated behaviors, such as physical activity, diet, and smoking, might provide more information when considered collectively than individually. Clinical trials specifically designed to investigate the influence of sociodemographic, clinical, behavioral, or molecular factors on the response to T2D preventive strategies are needed to generate valid and robust evidence before the implementation of T2D precision prevention strategies.

One area of promise warranting further research is the presence of prediabetes at baseline and whether this may be targeted in future precision prevention research. Low certainty evidence suggests that individuals at risk of T2D or with prediabetes at baseline benefit slightly more from prevention interventions than those not at risk of T2D[3–6]. However, the evidence is inconsistent, even though the studies report that a lifestyle intervention, compared to standard care, results in higher T2D reduction rates among studies conducted in Asia[17–20]. Beyond the methodological limitations of the available evidence, an additional reason for inconsistent evidence supporting the greater effectiveness of lifestyle interventions for the prevention of T2D among individuals with prediabetes is due to the heterogeneity that characterizes this condition. Prediabetes refers to a pathophysiological state of early alterations in glucose metabolism that precedes the development of diabetes. Still, the mechanisms by which glucose is elevated are very different and could range from those with primary alterations in insulin secretion pathways to those with primary insulin resistance[21]. Clinical trials specifically designed to capture the nuances and complexity of early glycemic alterations and whether individuals with distinct pathophysiological features benefit from more targeted preventive interventions are needed to fill the gap in current T2D precision prevention evidence.

Even though there are far more lifestyle intervention trials for the prevention of T2D than diet alone and diet supplementation trials, collectively, however, results for effect modification by any one factor are sparsely reported or arising from an evidence base of very different trials and patient populations. Further, many secondary analyses in this systematic review are derived from two single clinical interventions viz, the DPP and the DPS. Findings from available evidence contrast with recent clinical studies documenting variable responses to identical foods, diets, or lifestyle interventions based on inter-individual differences in demographic, clinical, genetic, gut microbiota, and lifestyle characteristics[22–24]. While these studies offer insights into variable postprandial metabolic response, their short follow-up periods, the lack of time-series data and changes in parameters that could influence response to interventions, and the inclusion of relatively young and healthy individuals preclude the generalizability to T2D prevention efforts. Whether the promise of T2D precision prevention is matched by evidence of the long-term beneficial impact remains uncertain. Still, interest and activity in this field are proliferating to identify factors underlying variable nutritional responses and develop algorithms to predict individual responses to nutrients, foods, and dietary patterns.

While recent studies support the benefits of losing body weight loss on the risk of developing T2D regardless of the mechanisms underlying T2D, there is still enormous variability in individual response to weight-loss interventions. For example, the DIET-FITS study[25], showed that weight change varied widely within

each study group, ranging from a loss of ~30 kg to a gain of ~10 kg. While weight loss is critical in T2D prevention, these findings reinforce the continued effort to identify molecular, environmental and social characteristics underlying the variable response to diabetes prevention interventions.

Our systematic review had some limitations. The scope of our literature review as part of the PDMI was broad and inclusive of diverse study designs, T2D prevention strategies, study populations, and effect modification analyses. Although this resulted in a heterogeneous evidence base and did not provide an opportunity for meta-analysis, we qualitatively synthesized the evidence for precision prevention. Our hypothesis originally spanned to include observational studies, which were ultimately excluded due to the uncertainty of their being readily related to clinical interventions. Protocol amendments were registered to reflect these decisions prior to study screening and extraction. Moreover, as our scope only included moderators of the intervention efficacy on T2D, which are typically measured prior to or at baseline[26], important mediators of the intervention effects on T2D as e.g., weight loss was not addressed and discussed. This will be important to address in future studies to gain a deeper understanding of heterogenous lifestyle interventions responses.

In conclusion, our systematic review and synthesis of the T2D prevention literature provide low to very low certainty evidence that sociodemographic, clinical, lifestyle, or molecular factors are more useful, valid, and consistent in informing T2D precision prevention strategies than current interventions. We also uncover several areas of potential for growth in the precision medicine field, including prospectively designed interventions and clinical trials incorporating the investigation of treatment response heterogeneity.

## Data availability

This systematic review compiles data available in clinical studies. The PMIDs of included studies are available in Table 1. The study-specific numeric estimates for the effect modification has been given in Supplementary Data 1. The source data for Fig. 2 is provided in Supplementary Data 2. All other extracted data have been summarized in the figures and tables presented in the manuscript and are available from the corresponding author on reasonable request.

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

## Acknowledgements

We thank Hugo Fitipaldi, Esther González-Padilla, Alisha Sha, and Jiaxi Yang for attending some of the working group meetings and/or for reviewing some of the abstracts. The *Precision Medicine in Diabetes Initiative* (PMDI) was established in 2018 by the American Diabetes Association (ADA) in partnership with the European Association for the Study of Diabetes (EASD). The ADA/EASD PMDI includes global thought leaders in precision diabetes medicine who are working to address the burgeoning need for better diabetes prevention and care through precision medicine (Nolan et al.[13]). This Systematic Review is written on behalf of the ADA/EASD PMDI as part of a comprehensive evidence evaluation in support of the 2nd International Consensus Report on Precision Diabetes Medicine (Tobias et al.[12]). The ADA/EASD Precision Diabetes Medicine Initiative, within which this work was conducted, has received the following support: The Covidence license was funded by Lund University (Sweden), for which technical support was provided by Maria Björklund and Krister Aronsson (Faculty of Medicine Library, Lund University, Sweden). Administrative support was provided by Lund University (Malmö, Sweden), the University of Chicago (IL, USA), and the American Diabetes Association (Washington D.C., USA). The Novo Nordisk Foundation (Hellerup, Denmark) provided grant support for in-person writing group meetings (PI: L Phillipson, University of Chicago, IL). D.B. was supported through an Early Career Research grant (ECR/2017/000640) from Science and Engineering Research Board (SERB), India. J.M. was partially supported by funding from the American Diabetes Association (7-21-JDFM-005) and the National Institutes of Health (P30 DK40561 and UG1 HD107691). R.J.F.L. received support through NNF18CC0034900; NNF20OC0059313 (Laureate Award), and DNRF161 (Chair).

## Author contributions

D.B., R.W.M., S.L.F., J.S.P., P.W.F., ADA/EASD PMDI, D.K.T., J.M., V.M., and R.J.F.L. contributed to the conception and design of the research questions. D.B., R.W.M., V.S., M.N., H.P.M., C.C., S.L.F., M.G.F., J.S.P., M.R.L., D.K.T., J.M., V.M., and R.J.F.L. contributed to the study screening and data extraction. D.K.T. and J.M. did the quality assessment; D.B., R.W.M., V.S., M.N., S.L.F., M.G.F., J.S.P., M.R.L., D.K.T., J.M., V.M., and R.J.F.L. summarized and interpreted the data. D.B., J.M., and R.J.F.L. drafted the paper; D.K.T. and V.M. revised it substantively. All authors edited the manuscript and approved the final version.

## Competing interests

The authors declare the following competing interests: R.W.M. and P.W.F. are employees of the Novo Nordisk Foundation, a private philanthropic enterprise foundation. The opinions expressed in this article do not necessarily reflect the perspectives of the Novo Nordisk Foundation. V.M. has acted as consultant and speaker and received research or educational grants from Novo Nordisk, MSD, Eli Lilly, Novartis, Boehringer Ingelheim, Lifescan J&J, Sanofi-Aventis, Roche Diagnostics, Abbott, and several Indian pharmaceutical companies, including USV, Dr. Reddy's Laboratories, and Sun Pharma. None of the other authors have any conflicts of interest to declare.

## Additional information

[1]Madras Diabetes Research Foundation, Chennai, India. [2]Department of Pathology & Molecular Medicine, McMaster University, Hamilton, ON, Canada. [3]Population Health Research Institute, Hamilton, ON, Canada. [4]Department of Translational Medicine, Medical Science, Novo Nordisk Foundation, Tuborg Havnevej 19, 2900 Hellerup, Denmark. [5]Division of Preventive Medicine, Department of Medicine, Brigham and Women's Hospital and Harvard Medical School, Boston, MA, USA. [6]Novo Nordisk Foundation Center for Basic Metabolic Research, Faculty of Health and Medical Sciences, University of Copenhagen, Copenhagen, Denmark. [7]Department of Clinical Sciences, Genetic and Molecular Epidemiology Unit, Lund University, Skåne University Hospital Malmö, Malmö, Sweden. [8]Department of Epidemiology, Geisel School of Medicine at Dartmouth, Hanover, NH, USA. [9]Institute of Health System Science, Feinstein Institutes for Medical Research, Northwell Health, Manhasset, NY, USA. [10]Department of Nutrition, Harvard T.H. Chan School of Public Health, Boston, MA, USA. [11]Division of Epidemiology and Community Health, School of Public Health, University of Minnesota, Minneapolis, MN, USA. [12]Centre for Physical Activity Research, Rigshospitalet, Copenhagen, Denmark. [13]Institute for Sports and Clinical Biomechanics, University of Southern Denmark, Odense, Denmark. [14]Lund University Diabetes Centre, Department of Clinical Sciences, Lund University, Malmo, Sweden. [15]Oxford Centre for Diabetes, Endocrinology and Metabolism, Radcliffe Department of Medicine, University of Oxford, Oxford, UK. [16]Diabetes Unit, Endocrine Division, Massachusetts General Hospital, Boston, MA, USA. [17]Center for Genomic Medicine, Massachusetts General Hospital, Boston, MA, USA. [18]Dr. Mohan's Diabetes Specialities Centre, Chennai, India. [19]Charles Bronfman Institute for Personalized Medicine, Icahn School of Medicine at Mount Sinai, New York, NY, USA. [202]These authors contributed equally: Deirdre K. Tobias, Jordi Merino. [203]These authors jointly supervised this work: Viswanathan Mohan, Ruth J. F. Loos. *A list of authors and their affiliations appears at the end of the paper. ✉email: ruth.loos@sund.ku.dk

## ADA/EASD PMDI

Deirdre K. Tobias[10,20], Jordi Merino[6,16,17,202], Abrar Ahmad[21], Catherine Aiken[22,23], Jamie L. Benham[24], Dhanasekaran Bodhini[25], Amy L. Clark[26], Kevin Colclough[27], Rosa Corcoy[28,29,30], Sara J. Cromer[16,31,32], Daisy Duan[33], Jamie L. Felton[34,35,36], Ellen C. Francis[37], Pieter Gillard[38], Véronique Gingras[39,40], Romy Gaillard[41], Eram Haider[42], Alice Hughes[27], Jennifer M. Ikle[43,44], Laura M. Jacobsen[45], Anna R. Kahkoska[46], Jarno L. T. Kettunen[47,48,49], Raymond J. Kreienkamp[16,17,31,50], Lee-Ling Lim[51,52,53], Jonna M. E. Männistö[54,55], Robert Massey[42], Niamh-Maire Mclennan[56], Rachel G. Miller[57], Mario Luca Morieri[58,59], Jasper Most[60], Rochelle N. Naylor[61], Bige Ozkan[62,63], Kashyap Amratlal Patel[27], Scott J. Pilla[64,65], Katsiaryna Prystupa[66,67], Sridharan Raghavan[68,69], Mary R. Rooney[62,70], Martin Schön[66,67,71], Zhila Semnani-Azad[10], Magdalena Sevilla-Gonzalez[31,32,72], Pernille Svalastoga[73,74], Wubet Worku Takele[75], Claudia Ha-ting Tam[53,76,77], Anne Cathrine B. Thuesen[6], Mustafa Tosur[78,79,80], Amelia S. Wallace[62,70], Caroline C. Wang[70], Jessie J. Wong[81], Jennifer M. Yamamoto[82], Katherine Young[27], Chloé Amouyal[83,84], Mette K. Andersen[6], Maxine P. Bonham[85], Mingling Chen[86], Feifei Cheng[87], Tinashe Chikowore[32,88,89,90], Sian C. Chivers[91], Christoffer Clemmensen[6], Dana Dabelea[92], Adem Y. Dawed[42], Aaron J. Deutsch[17,31,32], Laura T. Dickens[93], Linda A. DiMeglio[34,35,36,94], Monika Dudenhöffer-Pfeifer[21], Carmella Evans-Molina[34,35,36,95], María Mercè Fernández-Balsells[96,97], Hugo Fitipaldi[21], Stephanie L. Fitzpatrick[98], Stephen E. Gitelman[99], Mark O. Goodarzi[100,101], Jessica A. Grieger[102,103], Marta Guasch-Ferré[10,104], Nahal Habibi[102,103], Torben Hansen[6], Chuiguo Huang[53,76], Arianna Harris-Kawano[34,35,36], Heba M. Ismail[34,35,36], Benjamin Hoag[105,106], Randi K. Johnson[107,108], Angus G. Jones[27,109], Robert W. Koivula[110], Aaron Leong[16,32,111], Gloria K. W. Leung[85], Ingrid M. Libman[112], Kai Liu[102], S. Alice Long[113], William L. Lowe Jr.[114], Robert W. Morton[2,3,4], Ayesha A. Motala[115], Suna Onengut-Gumuscu[116], James S. Pankow[11], Maleesa Pathirana[102,103], Sofia Pazmino[117], Dianna Perez[34,35,36], John R. Petrie[118], Camille E. Powe[16,31,32,119], Alejandra Quinteros[102], Rashmi Jain[120,121], Debashree Ray[70,122], Mathias Ried-Larsen[12,13], Zeb Saeed[123], Vanessa Santhakumar[20], Sarah Kanbour[64,124], Sudipa Sarkar[64], Gabriela S. F. Monaco[34,35,36], Denise M. Scholtens[125], Elizabeth Selvin[62,70], Wayne Huey-Herng Sheu[126,127,128], Cate Speake[129], Maggie A. Stanislawski[107], Nele Steenackers[117], Andrea K. Steck[130], Norbert Stefan[67,131,132], Julie Støy[133], Rachael Taylor[134], Sok Cin Tye[135,136], Gebresilasea Gendisha Ukke[75], Marzhan Urazbayeva[79,137], Bart Van der Schueren[117,138], Camille Vatier[139,140], John M. Wentworth[141,142,143], Wesley Hannah[144,145], Sara L. White[91,146], Gechang Yu[53,76], Yingchai Zhang[53,76], Shao J. Zhou[103,147], Jacques Beltrand[148,149], Michel Polak[148,149], Ingvild Aukrust[73,150], Elisa de Franco[27], Sarah E. Flanagan[27], Kristin A. Maloney[151], Andrew McGovern[27], Janne Molnes[73,150], Mariam Nakabuye[6], Pål Rasmus Njølstad[73,74], Hugo Pomares-

Millan[8,21], Michele Provenzano[152], Cécile Saint-Martin[153], Cuilin Zhang[154,155], Yeyi Zhu[156,157], Sungyoung Auh[158], Russell de Souza[3,159], Andrea J. Fawcett[160,161], Chandra Gruber[162], Eskedar Getie Mekonnen[163,164], Emily Mixter[165], Diana Sherifali[3,166], Robert H. Eckel[167], John J. Nolan[168,169], Louis H. Philipson[165], Rebecca J. Brown[158], Liana K. Billings[170,171], Kristen Boyle[92], Tina Costacou[57], John M. Dennis[27], Jose C. Florez[16,17,31,32], Anna L. Gloyn[43,44,172], Maria F. Gomez[21,173], Peter A. Gottlieb[130], Siri Atma W. Greeley[174], Kurt Griffin[121,175], Andrew T. Hattersley[27,108], Irl B. Hirsch[176], Marie-France Hivert[16,177,178], Korey K. Hood[81], Jami L. Josefson[160], Soo Heon Kwak[179], Lori M. Laffel[180], Siew S. Lim[75], Ruth J. F. Loos [6,19,203] ✉, Ronald C. W. Ma[53,76,77], Chantal Mathieu[38], Nestoras Mathioudakis[64], James B. Meigs[32,111,181], Shivani Misra[182,183], Viswanathan Mohan[184], Rinki Murphy[185,186,187], Richard Oram[27,109], Katharine R. Owen[110,188], Susan E. Ozanne[189], Ewan R. Pearson[42], Wei Perng[92], Toni I. Pollin[151,190], Rodica Pop-Busui[191], Richard E. Pratley[192], Leanne M. Redman[193], Maria J. Redondo[78,79], Rebecca M. Reynolds[56], Robert K. Semple[56,194], Jennifer L. Sherr[195], Emily K. Sims[34,35,36], Arianne Sweeting[196,197], Tiinamaija Tuomi[47,142,49], Miriam S. Udler[16,17,31,32], Kimberly K. Vesco[198], Tina Vilsbøll[199,200], Robert Wagner[66,67,201], Stephen S. Rich[116] & Paul W. Franks[4,10,21,110]

[20]Division of Preventative Medicine, Department of Medicine, Brigham and Women's Hospital and Harvard Medical School, Boston, MA, USA. [21]Department of Clinical Sciences, Lund University Diabetes Centre, Lund University, Malmö, Sweden. [22]Department of Obstetrics and Gynaecology, The Rosie Hospital, Cambridge, UK. [23]NIHR Cambridge Biomedical Research Centre, University of Cambridge, Cambridge, UK. [24]Departments of Medicine and Community Health Sciences, Cumming School of Medicine, University of Calgary, Calgary, AB, Canada. [25]Department of Molecular Genetics, Madras Diabetes Research Foundation, Chennai, India. [26]Division of Pediatric Endocrinology, Department of Pediatrics, Saint Louis University School of Medicine, SSM Health Cardinal Glennon Children's Hospital, St. Louis, MO, USA. [27]Department of Clinical and Biomedical Sciences, University of Exeter Medical School, Exeter, Devon, UK. [28]CIBER-BBN, ISCIII, Madrid, Spain. [29]Institut d'Investigació Biomèdica Sant Pau (IIB SANT PAU), Barcelona, Spain. [30]Departament de Medicina, Universitat Autònoma de Barcelona, Bellaterra, Spain. [31]Programs in Metabolism and Medical & Population Genetics, Broad Institute, Cambridge, MA, USA. [32]Department of Medicine, Harvard Medical School, Boston, MA, USA. [33]Division of Endocrinology, Diabetes and Metabolism, Johns Hopkins University School of Medicine, Baltimore, MD, USA. [34]Department of Pediatrics, Indiana University School of Medicine, Indianapolis, IN, USA. [35]Herman B Wells Center for Pediatric Research, Indiana University School of Medicine, Indianapolis, IN, USA. [36]Center for Diabetes and Metabolic Diseases, Indiana University School of Medicine, Indianapolis, IN, USA. [37]Department of Biostatistics and Epidemiology, Rutgers School of Public Health, Piscataway, NJ, USA. [38]University Hospital Leuven, Leuven, Belgium. [39]Department of Nutrition, Université de Montréal, Montreal, QC, Canada. [40]Research Center, Sainte-Justine University Hospital Center, Montreal, QC, Canada. [41]Department of Pediatrics, Erasmus Medical Center, Rotterdam, The Netherlands. [42]Division of Population Health & Genomics, School of Medicine, University of Dundee, Dundee, UK. [43]Department of Pediatrics, Stanford School of Medicine, Stanford University, Stanford, CA, USA. [44]Stanford Diabetes Research Center, Stanford School of Medicine, Stanford University, Stanford, CA, USA. [45]University of Florida, Gainesville, FL, USA. [46]Department of Nutrition, University of North Carolina at Chapel Hill, Chapel Hill, NC, USA. [47]Helsinki University Hospital, Abdominal Centre/Endocrinology, Helsinki, Finland. [48]Folkhalsan Research Center, Helsinki, Finland. [49]Institute for Molecular Medicine Finland FIMM, University of Helsinki, Helsinki, Finland. [50]Department of Pediatrics, Division of Endocrinology, Boston Children's Hospital, Boston, MA, USA. [51]Department of Medicine, Faculty of Medicine, University of Malaya, Kuala Lumpur, Malaysia. [52]Asia Diabetes Foundation, Hong Kong SAR, China. [53]Department of Medicine & Therapeutics, Chinese University of Hong Kong, Hong Kong, Hong Kong SAR, China. [54]Departments of Pediatrics and Clinical Genetics, Kuopio University Hospital, Kuopio, Finland. [55]Department of Medicine, University of Eastern Finland, Kuopio, Finland. [56]Centre for Cardiovascular Science, Queen's Medical Research Institute, University of Edinburgh, Edinburgh, UK. [57]Department of Epidemiology, University of Pittsburgh, Pittsburgh, PA, USA. [58]Metabolic Disease Unit, University Hospital of Padova, Padova, Italy. [59]Department of Medicine, University of Padova, Padova, Italy. [60]Department of Orthopedics, Zuyderland Medical Center, Sittard-Geleen, The Netherlands. [61]Departments of Pediatrics and Medicine, University of Chicago, Chicago, IL, USA. [62]Welch Center for Prevention, Epidemiology, and Clinical Research, Johns Hopkins Bloomberg School of Public Health, Baltimore, MD, USA. [63]Ciccarone Center for the Prevention of Cardiovascular Disease, Johns Hopkins School of Medicine, Baltimore, MD, USA. [64]Department of Medicine, Johns Hopkins University, Baltimore, MD, USA. [65]Department of Health Policy and Management, Johns Hopkins University Bloomberg School of Public Health, Baltimore, MD, USA. [66]Institute for Clinical Diabetology, German Diabetes Center, Leibniz Center for Diabetes Research at Heinrich Heine University Düsseldorf, Auf'm Hennekamp 65, 40225 Düsseldorf, Germany. [67]German Center for Diabetes Research (DZD), Ingolstädter Landstraße 1, 85764 Neuherberg, Germany. [68]Section of Academic Primary Care, US Department of Veterans Affairs Eastern Colorado Health Care System, Aurora, CO, USA. [69]Department of Medicine, University of Colorado School of Medicine, Aurora, CO, USA. [70]Department of Epidemiology, Johns Hopkins Bloomberg School of Public Health, Baltimore, MD, USA. [71]Institute of Experimental Endocrinology, Biomedical Research Center, Slovak Academy of Sciences, Bratislava, Slovakia. [72]Clinical and Translational Epidemiology Unit, Massachusetts General Hospital, Boston, MA, USA. [73]Mohn Center for Diabetes Precision Medicine, Department of Clinical Science, University of Bergen, Bergen, Norway. [74]Children and Youth Clinic, Haukeland University Hospital, Bergen, Norway. [75]Eastern Health Clinical School, Monash University, Melbourne, VIC, Australia. [76]Laboratory for Molecular Epidemiology in Diabetes, Li Ka Shing Institute of Health Sciences, The Chinese University of Hong Kong, Hong Kong, China. [77]Hong Kong Institute of Diabetes and Obesity, The Chinese University of Hong Kong, Hong Kong, China. [78]Department of Pediatrics, Baylor College of Medicine, Houston, TX, USA. [79]Division of Pediatric Diabetes and Endocrinology, Texas Children's Hospital, Houston, TX, USA. [80]Children's Nutrition Research Center, USDA/ARS, Houston, TX, USA. [81]Stanford University School of Medicine, Stanford, CA, USA. [82]Internal Medicine, University of Manitoba, Winnipeg, MB, Canada. [83]Department of Diabetology, APHP, Paris, France. [84]Sorbonne Université, INSERM, NutriOmic Team, Paris, France. [85]Department of Nutrition, Dietetics and Food, Monash University, Melbourne, VIC, Australia. [86]Monash Centre for Health Research and Implementation, Monash University, Clayton, VIC, Australia. [87]Health Management Center, The Second Affiliated Hospital of

Chongqing Medical University, Chongqing Medical University, Chongqing, China. [88]MRC/Wits Developmental Pathways for Health Research Unit, Department of Paediatrics, Faculty of Health Sciences, University of the Witwatersrand, Johannesburg, South Africa. [89]Channing Division of Network Medicine, Brigham and Women's Hospital, Boston, MA, USA. [90]Sydney Brenner Institute for Molecular Bioscience, Faculty of Health Sciences, University of the Witwatersrand, Johannesburg, South Africa. [91]Department of Women and Children's health, King's College London, London, UK. [92]Lifecourse Epidemiology of Adiposity and Diabetes (LEAD) Center, University of Colorado Anschutz Medical Campus, Aurora, CO, USA. [93]Section of Adult and Pediatric Endocrinology, Diabetes and Metabolism, Kovler Diabetes Center, University of Chicago, Chicago, IL, USA. [94]Department of Pediatrics, Riley Hospital for Children, Indiana University School of Medicine, Indianapolis, IN, USA. [95]Richard L. Roudebush VAMC, Indianapolis, IN, USA. [96]Biomedical Research Institute Girona, IdIBGi, Girona, Spain. [97]Diabetes, Endocrinology and Nutrition Unit Girona, University Hospital Dr Josep Trueta, Girona, Spain. [98]Institute of Health System Science, Feinstein Institutes for Medical Research, Northwell Health, Manhasset, NY, USA. [99]University of California at San Francisco, Department of Pediatrics, Diabetes Center, San Francisco, CA, USA. [100]Division of Endocrinology, Diabetes and Metabolism, Cedars-Sinai Medical Center, Los Angeles, CA, USA. [101]Department of Medicine, Cedars-Sinai Medical Center, Los Angeles, CA, USA. [102]Adelaide Medical School, Faculty of Health and Medical Sciences, The University of Adelaide, Adelaide, SA, Australia. [103]Robinson Research Institute, The University of Adelaide, Adelaide, SA, Australia. [104]Department of Public Health and Novo Nordisk Foundation Center for Basic Metabolic Research, Faculty of Health and Medical Sciences, University of Copenhagen, 1014 Copenhagen, Denmark. [105]Division of Endocrinology and Diabetes, Department of Pediatrics, Sanford Children's Hospital, Sioux Falls, SD, USA. [106]University of South Dakota School of Medicine, E Clark St, Vermillion, SD, USA. [107]Department of Biomedical Informatics, University of Colorado Anschutz Medical Campus, Aurora, CO, USA. [108]Department of Epidemiology, Colorado School of Public Health, Aurora, CO, USA. [109]Royal Devon University Healthcare NHS Foundation Trust, Exeter, UK. [110]Oxford Centre for Diabetes, Endocrinology and Metabolism, University of Oxford, Oxford, UK. [111]Division of General Internal Medicine, Massachusetts General Hospital, Boston, MA, USA. [112]UPMC Children's Hospital of Pittsburgh, Pittsburgh, PA, USA. [113]Center for Translational Immunology, Benaroya Research Institute, Seattle, WA, USA. [114]Department of Medicine, Northwestern University Feinberg School of Medicine, Chicago, IL, USA. [115]Department of Diabetes and Endocrinology, Nelson R Mandela School of Medicine, University of KwaZulu-Natal, Durban, South Africa. [116]Center for Public Health Genomics, Department of Public Health Sciences, University of Virginia, Charlottesville, VA, USA. [117]Department of Chronic Diseases and Metabolism, Clinical and Experimental Endocrinology, KU Leuven, Leuven, Belgium. [118]School of Health and Wellbeing, College of Medical, Veterinary and Life Sciences, University of Glasgow, Glasgow, UK. [119]Department of Obstetrics, Gynecology, and Reproductive Biology, Massachusetts General Hospital and Harvard Medical School, Boston, MA, USA. [120]Sanford Children's Specialty Clinic, Sioux Falls, SD, USA. [121]Department of Pediatrics, Sanford School of Medicine, University of South Dakota, Sioux Falls, SD, USA. [122]Department of Biostatistics, Johns Hopkins Bloomberg School of Public Health, Baltimore, MD, USA. [123]Department of Medicine, Division of Endocrinology, Diabetes and Metabolism, Indiana University School of Medicine, Indianapolis, IN, USA. [124]AMAN Hospital, Doha, Qatar. [125]Department of Preventive Medicine, Division of Biostatistics, Northwestern University Feinberg School of Medicine, Chicago, IL, USA. [126]Institute of Molecular and Genomic Medicine, National Health Research Institutes, Taipei City, Taiwan. [127]Division of Endocrinology and Metabolism, Taichung Veterans General Hospital, Taichung, Taiwan. [128]Division of Endocrinology and Metabolism, Taipei Veterans General Hospital, Taipei, Taiwan. [129]Center for Interventional Immunology, Benaroya Research Institute, Seattle, WA, USA. [130]Barbara Davis Center for Diabetes, University of Colorado Anschutz Medical Campus, Aurora, CO, USA. [131]University Hospital of Tübingen, Tübingen, Germany. [132]Institute of Diabetes Research and Metabolic Diseases (IDM), Helmholtz Center Munich, Neuherberg, Germany. [133]Steno Diabetes Center Aarhus, Aarhus University Hospital, Aarhus, Denmark. [134]University of Newcastle, Newcastle upon Tyne, UK. [135]Sections on Genetics and Epidemiology, Joslin Diabetes Center, Harvard Medical School, Boston, MA, USA. [136]Department of Clinical Pharmacy and Pharmacology, University Medical Center Groningen, Groningen, The Netherlands. [137]Gastroenterology, Baylor College of Medicine, Houston, TX, USA. [138]Department of Endocrinology, University Hospitals Leuven, Leuven, Belgium. [139]Sorbonne University, Inserm U938, Saint-Antoine Research Centre, Institute of Cardiometabolism and Nutrition, 75012 Paris, France. [140]Department of Endocrinology, Diabetology and Reproductive Endocrinology, Assistance Publique-Hôpitaux de Paris, Saint-Antoine University Hospital, National Reference Center for Rare Diseases of Insulin Secretion and Insulin Sensitivity (PRISIS), Paris, France. [141]Royal Melbourne Hospital Department of Diabetes and Endocrinology, Parkville, VIC, Australia. [142]Walter and Eliza Hall Institute, Parkville, VIC, Australia. [143]University of Melbourne Department of Medicine, Parkville, VIC, Australia. [144]Deakin University, Melbourne, VIC, Australia. [145]Department of Epidemiology, Madras Diabetes Research Foundation, Chennai, India. [146]Department of Diabetes and Endocrinology, Guy's and St Thomas' Hospitals NHS Foundation Trust, London, UK. [147]School of Agriculture, Food and Wine, University of Adelaide, Adelaide, SA, Australia. [148]Institut Cochin, Inserm U, 10116 Paris, France. [149]Pediatric Endocrinology and Diabetes, Hopital Necker Enfants Malades, APHP Centre, Université de Paris, Paris, France. [150]Department of Medical Genetics, Haukeland University Hospital, Bergen, Norway. [151]Department of Medicine, University of Maryland School of Medicine, Baltimore, MD, USA. [152]Nephrology, Dialysis and Renal Transplant Unit, IRCCS—Azienda Ospedaliero-Universitaria di Bologna, Alma Mater Studiorum University of Bologna, Bologna, Italy. [153]Department of Medical Genetics, AP-HP Pitié-Salpêtrière Hospital, Sorbonne University, Paris, France. [154]Global Center for Asian Women's Health, Yong Loo Lin School of Medicine, National University of Singapore, Singapore, Singapore. [155]Department of Obstetrics and Gynecology, Yong Loo Lin School of Medicine, National University of Singapore, Singapore, Singapore. [156]Kaiser Permanente Northern California Division of Research, Oakland, CA, USA. [157]Department of Epidemiology and Biostatistics, University of California San Francisco, San Francisco, CA, USA. [158]National Institute of Diabetes and Digestive and Kidney Diseases, National Institutes of Health, Bethesda, MD, USA. [159]Department of Health Research Methods, Evidence, and Impact, Faculty of Health Sciences, McMaster University, Hamilton, ON, Canada. [160]Ann & Robert H. Lurie Children's Hospital of Chicago, Department of Pediatrics, Northwestern University Feinberg School of Medicine, Chicago, IL, USA. [161]Department of Clinical and Organizational Development, Chicago, IL, USA. [162]American Diabetes Association, Arlington, VA, USA. [163]College of Medicine and Health Sciences, University of Gondar, Gondar, Ethiopia. [164]Global Health Institute, Faculty of Medicine and Health Sciences, University of Antwerp, 2160 Antwerp, Belgium. [165]Department of Medicine and Kovler Diabetes Center, University of Chicago, Chicago, IL, USA. [166]School of Nursing, Faculty of Health Sciences, McMaster University, Hamilton, ON, Canada. [167]Division of Endocrinology, Metabolism, Diabetes, University of Colorado, Boulder, CO, USA. [168]Department of Clinical Medicine, School of Medicine, Trinity College Dublin, Dublin, Ireland. [169]Department of Endocrinology, Wexford General Hospital, Wexford, Ireland. [170]Division of Endocrinology, NorthShore University HealthSystem, Skokie, IL, USA. [171]Department of Medicine, Prtizker School of Medicine, University of Chicago, Chicago, IL, USA. [172]Department of Genetics, Stanford School of Medicine, Stanford University, Stanford, CA, USA. [173]Faculty of Health, Aarhus University, Aarhus, Denmark. [174]Departments of Pediatrics and Medicine and Kovler Diabetes Center, University of Chicago, Chicago, IL, USA. [175]Sanford Research, Sioux Falls, SD, USA. [176]University of Washington School of Medicine, Seattle, WA, USA. [177]Department of Population Medicine, Harvard Medical School, Harvard Pilgrim Health Care Institute, Boston, MA, USA. [178]Department of Medicine, Universite de Sherbrooke, Sherbrooke, QC, Canada. [179]Department of Internal Medicine, Seoul National University College of Medicine, Seoul National University Hospital, Seoul, Republic of Korea. [180]Joslin Diabetes Center, Harvard Medical School, Boston, MA, USA. [181]Broad Institute, Cambridge, MA, USA. [182]Division of Metabolism, Digestion and Reproduction, Imperial College London, London, UK. [183]Department of Diabetes & Endocrinology, Imperial College Healthcare NHS Trust, London, UK. [184]Department of Diabetology, Madras Diabetes Research Foundation & Dr. Mohan's Diabetes

Specialities Centre, Chennai, India. [185]Department of Medicine, Faculty of Medicine and Health Sciences, University of Auckland, Auckland, New Zealand. [186]Auckland Diabetes Centre, Te Whatu Ora Health New Zealand, Auckland, New Zealand. [187]Medical Bariatric Service, Te Whatu Ora Counties, Health New Zealand, Auckland, New Zealand. [188]Oxford NIHR Biomedical Research Centre, University of Oxford, Oxford, UK. [189]University of Cambridge, Metabolic Research Laboratories and MRC Metabolic Diseases Unit, Wellcome-MRC Institute of Metabolic Science, Cambridge, UK. [190]Department of Epidemiology & Public Health, University of Maryland School of Medicine, Baltimore, MD, USA. [191]Department of Internal Medicine, Division of Metabolism, Endocrinology and Diabetes, University of Michigan, Ann Arbor, MI, USA. [192]AdventHealth Translational Research Institute, Orlando, FL, USA. [193]Pennington Biomedical Research Center, Baton Rouge, LA, USA. [194]MRC Human Genetics Unit, Institute of Genetics and Cancer, University of Edinburgh, Edinburgh, UK. [195]Yale School of Medicine, New Haven, CT, USA. [196]Faculty of Medicine and Health, University of Sydney, Sydney, NSW, Australia. [197]Department of Endocrinology, Royal Prince Alfred Hospital, Sydney, NSW, Australia. [198]Kaiser Permanente Northwest, Kaiser Permanente Center for Health Research, Portland, OR, USA. [199]Clinial Research, Steno Diabetes Center Copenhagen, Herlev, Denmark. [200]Department of Clinical Medicine, Faculty of Health and Medical Sciences, University of Copenhagen, Copenhagen, Denmark. [201]Department of Endocrinology and Diabetology, University Hospital Düsseldorf, Heinrich Heine University Düsseldorf, Moorenstr. 5, 40225 Düsseldorf, Germany.

