## [Peer Review File · Communications Medicine]

Reviewers' comments:

Reviewer #1 (Remarks to the Author):

It is generally accepted view that type 2 diabetes (T2D) is preventable, or its onset may be delayed by lifestyle changes, and the strongest evidence comes from randomized clinical trials carried out with prediabetic individuals with impaired glucose tolerance or impaired fasting glucose at baseline. The current review is based on systematic literature search and is written on behalf of ADA/EASD Precision Medicine Initiative. The main question is whether different socio-economic, lifestyle, behavior, body composition and molecular/genetic factors modify the success of preventive effect regarding the risk of T2D. Altogether 80 publications with quite different quality were included in the review process. In summary, the evidence that the factors examined, including genetic risk markers, could modify the intervention effect was in general low or very low. Nevertheless, the authors found a low certainty of evidence to support conclusions that those with poorer health status, particularly those with prediabetes, may benefit more from T2D prevention strategies compared to healthier counterparts.

Comments

My main comment concerns the selection of studies in analyses and why including quite different studies in analyses. I think that the strongest evidence comes from high quality randomized trials primarily aimed to prevent T2D by changing lifestyles. The authors should clarify why they have included non-randomized studies and singly arms interventions in analyses and what is the additional evidence coming from observational studies. In my mind, these studies are highly heterogenous to make any definite conclusions. Study populations are very variable in terms of number of individuals included (302 -48 835), ethnicity and time of follow-up (one visit to 10 years follow-up). Next, the intervention effect is highly dependent on the success of a given lifestyle intervention applied, and especially the magnitude of weight loss plays central role in the prevention of T2D in obese prediabetic individuals, as shown e.g., by Penn L et al (PLoS One 2013). I suggest that the authors should make clear difference between the evidence that is coming from randomized trials aimed at prevention of T2D, the evidence coming from secondary analyses of other studies included in this analysis (e.g PREDIMET) and then other evidence from non-randomized trials. What I am saying is that the quality and study design of the studies may matter a lot. Nevertheless, the main conclusions made by the authors sound correct in many instances, but as for the importance of weight loss in the prevention (intervention trials) and even remission of T2D (e.g., DiRECT studies), the evidence is strong. I think DiRECT findings support the importance of weight loss. Furthermore, a recent meta-analysis of healthy dietary patterns/lifestyles also suggests that they may have marked impact on the prevention of the disease (Khan A et al. Diabetes Care 2023).

Specific comments

Line 62, Nat Med, year is missing.

Methods, study design. The authors should clarify why quite different study designs were accepted into this analysis. Furthermore, there are many meta-analyses done earlier based on well-controlled lifestyle interventions regarding the prevention of T2D. Why were they not utilized in this analysis? Lines 170-173 needs clarifications, did studies with different study designs have different impacts or were they analyzed otherwise separately? What does it mean "collective evidence" in this regard? Note that meta-analyses were not carried out.

Lines 181-191. Did the location of the study centre or race/ethnicity influence the results.

Furthermore, Asian people are leaner than e.g., those of European origin, did this fact have any effect on the results?

Lines 197-198 deals with high vs. low fat diets. Regarding PREDIMET study results, the control diet

was rather high fat diet, as well (37 E%), see the resp. supplemental table 9 of this PREDIMET publication (New Engl J Med 2018). In fact, the energy % difference was only 4 % (37 vs. 41 %) between the control and intervention diets. Therefore, if the main conclusion on high vs. low fat diets is mainly based on this study, it may not be correct. See also the abstract, lines 205-206 and conclusions, and Table 1 as well! Furthermore, most controlled intervention trials in the prevention of T2D have been carried out with rather low-fat diets.

Lines 208-210, If there are concerns with some intervention trials why then they were included in analyses? I refer to many meta-analyses applying GRADE on the subject. Regarding lifestyle interventions, a priori they may not be blind in terms of the intervention arms. The authors may give more information in this regard in the text.

Lines 224- gives the results on the effect of initial BMI on weight loss success but does not give any analysis on the degree of body weight loss on the outcome measure i.e., incident diabetes. In this regard, the conclusions may be different as I pointed out above.

Regarding the effect of genetic and other molecular factors, the evidence is scares and further controlled trials are needed with appropriate study designs as the authors also pointed out.

Lines 263-270, may be removed or shortened because of lacking confirmation?

Lines 282- Discussion, even if the initial BMI does not play a role in the success, I think that the authors should comment on the importance of weight reduction in the prevention of T2D. The readers may misunderstand the message in the first paragraph in discussion.

Lines 293-294, as said above, it is weight loss that matters along with healthy diet and lifestyles like physical activity in the prevention of T2D.

Lines 315-317, I do not totally understand the message of this sentence. Do the authors challenge the evidence of major T2D prevention trials or not? I think we cannot compare the studies carried out with high risk people to those with normal glucose tolerance since these two groups are quite different in terms of the absolute and relative risk of T2D.

Ref. 52, there are similar data on the DPS study population regarding TCF7L2 polymorphism, Wang J et al. Diabetologia 2007.

Discussion, lines 319-329. My view is that although based on limited data published, current evidence suggests that lifestyles matter regardless of heterogenous matter of the disease and that insulin resistance is highly modifiable by weight loss and increased physical activity, and furthermore, very limited evidence suggests that weight loss may also help to maintain insulin secretion capacity or even improve it, see e.g., DiRECT study results and those published from DPS (de Mello V et al. Diabetes Care 2013), and Indian Study (Snehalatha C et al. Diabetes Care 2009) and multiple DPP publications related to the topic.

Finally, I totally agree with the authors regarding the main conclusions of the manuscript.

.

Reviewer #2 (Remarks to the Author):

Prevention of type 2 diabetes is a key public health issue. The initiative outlined here to investigate intervention efficacy based on different study and participant characteristics is of crucial importance. The rationale for this manuscript is therefore reasonable. This is a well written manuscript with generally appropriate methodology, analysis and interpretation.

It would benefit from more details with regards to study characteristics to allow further interpretation.

Line 81 – A comment on the relatively efficacy of diet or physical activity components on prevention of diabetes (both in association and not in association with weight loss) would be useful here.

Line 90 – A comment here on the use (or not) on implementation science in aiding translation and scalability in these interventions would be useful here.

The search is reasonably old and I would recommend an update be conducted.

Methods: The term lifestyle for describing the intervention is used frequently, and then often followed by diet/behavioural etc. I would suggest an upfront definition what constitutes a lifestyle intervention – ie diet, PA? Psychological interventions? This is currently unclear, it is done on line 125 to some extent but I would move this statement upfront in the methods and then just refer to lifestyle after this. Similarly for line 163 I would define this in more detail (ie lifestyle (and what exactly is meant by this, ie multicomponent), diet alone, supplement etc). In table 2 it is also referred to as 'intensive' lifestyle – I would define and justify this term (eg is this based on total session number or another factor)? For example, smoking is referred to for the first time on line 193. And confirming there were no studies for exercise in isolation?

Please define and reference the certainty of evidence system in the methods

Provide a reference for the justification of including studies with $n > 100$

Line 125 - I would describe in greater detail what type of effect modification data you accepted here. ie statistical interactions? Stratified data? Heterogeneity in response? Any of these or others? The language in the results should then reflect the methods used

Suggest reformat Table 3 to be consistent with the Table 1 subheadings of lifestyle/diet/supplement and add in study methodology (RCT or not) and if weight loss was a specific aim of the study or not.

I would think a meta-analysis may be possible for some of the subgroup analysis in the Lifestyle interventions as the sample size for number of studies is reasonable (at least for age?)

Line 206 – Please add details on what type of high fat diet

Line 209 – generally low quality? Or generally low risk of bias? Unclear

Line 210 – Blinding is very difficult in lifestyle interventions and I would comment on this. Can you also comment more on if this was blinding of objective or subjective outcome measurement?

Line 217 – 'Evidence presented in studies investigating the effect of a lifestyle intervention according to differences in sociodemographic and clinical characteristics did not indicate statistically different effects for age, sex, race/ethnicity, or socioeconomic status.' This is unclear as some of the studies in Table 2 had effect modification.

Line 224 onwards - Additional supplemental data for numeric data for effect modification data is required including numeric results and the methodology for comparing differences between groups.

Line 212 – Which concerns for study design?

Line 335 – Which two studies?

Did the studies with clinical effect modifiers comment on if these were independent of BMI (eg prediabetes, age etc)

With regards to consideration of underlying clinical conditions, this is particularly relevant given the finding of some suggestion of a modifying effect of prediabetes. This should be commented on further for specific conditions (eg GDM or others such as PCOS, other pregnancy complications etc). I also note the GDM exclusion criteria, please provide the relevant protocol registration details

Table 1 would benefit from a column on if weight loss was an explicit aim of the study

RESPONSE TO REVIEWERS

Reviewer #1:

It is generally accepted view that type 2 diabetes (T2D) is preventable, or its onset may be delayed by lifestyle changes, and the strongest evidence comes from randomized clinical trials carried out with prediabetic individuals with impaired glucose tolerance or impaired fasting glucose at baseline. The current review is based on systematic literature search and is written on behalf of ADA/EASD Precision Medicine Initiative. The main question is whether different socio-economic, lifestyle, behavior, body composition and molecular/genetic factors modify the success of preventive effect regarding the risk of T2D. Altogether 80 publications with quite different quality were included in the review process. In summary, the evidence that the factors examined, including genetic risk markers, could modify the intervention effect was in general low or very low. Nevertheless, the authors found a low certainty of evidence to support conclusions that those with poorer health status, particularly those with prediabetes, may benefit more from T2D prevention strategies compared to healthier counterparts.

Response: We thank the Reviewer for taking the time to read our manuscript and provide detailed comments. A point-by-point response to all the comments has been given below.

Comments

My main comment concerns the selection of studies in analyses and why including quite different studies in analyses. I think that the strongest evidence comes from high quality randomized trials primarily aimed to prevent T2D by changing lifestyles. The authors should clarify why they have included non-randomized studies and singly arms interventions in analyses and what is the additional evidence coming from observational studies. In my mind, these studies are highly heterogenous to make any definite conclusions. Study populations are very variable in terms of number of individuals included (302 - 48 835), ethnicity and time of follow-up (one visit to 10 years follow-up). Next, the intervention effect is highly dependent on the success of a given lifestyle intervention applied, and especially the magnitude of weight loss plays central role in the prevention of T2D in obese prediabetic individuals, as shown e.g., by Penn L et al (PLoS One 2013). I suggest that the authors should make clear difference between the evidence that is coming from randomized trials aimed at prevention of T2D, the evidence coming from secondary analyses of other studies included in this analysis (e,g PREDIMET) and then other evidence from non-randomized trials. What I am saying is that the quality and study design of the studies may matter a lot. Nevertheless, the main conclusions made by the

authors sound correct in many instances, but as for the importance of weight loss in the prevention (intervention trials) and even remission of T2D (e.g., DiRECT studies), the evidence is strong. I think DiRECT findings support the importance of weight loss. Furthermore, a recent meta-analysis of healthy dietary patterns/lifestyles also suggests that they may have marked impact on the prevention of the disease (Khan A et al. Diabetes Care 2023).

Response: We agree with the Reviewer that high-quality randomized trials primarily designed to study the effect of behavioral interventions to prevent T2D provide the strongest evidence. However, in our systematic review, our focus is not on the effect of the interventions (i.e. main effects) as such but on the evidence that this effect is modified by sociodemographic, clinical, behavioral and molecular factors. While the vast majority of studies (N=76 or 94%) included in our review are RCTs to study the main effects of intervention, we found no RCTs designed specifically for studying effect modification/ differential intervention effects. Thus, studies that examine the effect modification perform secondary analyses using the data available from RCTs. Though these trials are randomized to study the main effect, the randomization block is not conserved to study the differential intervention effects. For instance, the DPP is a randomized controlled trial. However, when studying whether baseline fasting glucose levels affect the success of the interventions, the DPP cannot be considered an RCT as participants were not randomized based on baseline glucose levels. Thus, since none of the studies included in our review are RCTs to study the effect modification, including the three non-randomized trials is unlikely to affect the overall conclusions and certainty of evidence. We have clarified this point in the Methods section.

Page 8, line 139: *The majority of studies (N =76 or 94%) included in this review are RCTs to examine the effect of the intervention on T2D incidence. However, as our focus is on the modification of the intervention effect by sociodemographic, clinical, behavioral, and molecular factors, none of these trials can be considered randomized for the purpose of this review, as the randomization block is not conserved.*

We agree that the prevention of weight gain and weight loss as a consequence of an intervention plays an important role in the prevention of and for remission of T2D. In this context, weight change would be considered an important mediator of the effects of lifestyle intervention on T2D. The primary objective of this systematic review includes only potential moderators of the intervention effects. Investigating effect modification is typically done only using population characteristics assessed prior to or at the baseline measurements (see <https://doi.org/10.1016/j.jclinepi.2021.03.009>). While we agree that investigations specifically designed to tackle the potential modifiers of weight loss are needed, it is beyond the scope to include weight changes in our analyses. We have added the following information to the manuscript.

Page 18 line 373: *Moreover, as our scope only included moderators of the intervention efficacy on T2D, which are typically measured prior to or at baseline²⁵, important mediators of the intervention effects on T2D as e.g. weight loss, was not addressed and discussed. This will be important to address in future studies to gain a deeper understanding of heterogenous lifestyle interventions responses.*

Specific Comments

1. Line 62, Nat Med, year is missing.

Response: We have amended this.

2. Methods, study design. The authors should clarify why quite different study designs were accepted into this analysis. Furthermore, there are many meta-analyses done earlier based on well-controlled lifestyle interventions regarding the prevention of T2D. Why were they not utilized in this analysis?

Response: We agree with the Reviewer that there are many meta-analyses of RCTs investigating main effects of intervention. However, our main purpose was to assess whether this effect varies according to individual characteristics. To our knowledge, this is the first systematic review examining moderators of intervention effects. As described above, no randomized studies have been designed to specifically investigate the effect of a secondary variable on the efficacy of T2D preventive interventions. In addition to the randomization issues highlighted before, these studies suffer from statistical power (i.e., studies not designed for stratification purposes), confounding, and multiple testing.

3. Lines 170-173 needs clarifications, did studies with different study designs have different impacts or were they analyzed otherwise separately? What does it mean “collective evidence” in this regard? Note that meta-analyses were not carried out.

Response: We would like to clarify that a meta-analysis was not feasible due to marked differences in studies included, and we qualitatively synthesized the evidence for each modifier based on the direction of findings reported in the studies.

By “collective evidence,” we meant the certainty of the evidence for each modifier based on the criteria established by the Diabetes Canada Clinical Practice Scale to assess the certainty of the evidence. We have added the following information:

Page 9, line 180: *We qualitatively synthesized the evidence for each modifier based on the direction of findings reported in available studies. We used the Diabetes Canada Clinical Practice Scale to assess the certainty of the evidence for a given effect modifier.*

4. Lines 181-191. Did the location of the study centre or race/ethnicity influence the results. Furthermore, Asian people are leaner than e.g., those of European origin, did this fact have any effect on the results?

Response: Thanks for this relevant suggestion. Actually, the IDPP and DCLIP trials conducted on Asians showed a lesser effect, 28-32% reduction in diabetes incidence in intervention group as compared to 58% in DPP and DPS trials. However, due to the marked differences between these trials in terms of sample size, methodology, intensity of intervention etc, it may not be appropriate to compare them to find if ethnicity modified the effect of intervention. Instead, it would be relevant to discuss the studies that have reported the intervention effect stratified based on location/ethnicity. There were three studies examining the role of geographical location on the efficacy of T2D preventive interventions. These studies showed no evidence of an interaction between geographical location and lifestyle or dietary supplement interventions on the risk of T2D (Table 2). Studies stratified by race/ethnicity showed no evidence that race/ethnicity modified the efficacy of dietary and lifestyle interventions to prevent T2D. The quality of evidence ranged from low to very-low certainty. We have added the following information:

Page 11, line 231: *Certainty of evidence to indicate different effects for sociodemographic characteristics such as age, sex, race/ethnicity, socioeconomic status or geographic location in response to lifestyle intervention was low.*

5. Lines 197-198 deals with high vs. low fat diets. Regarding PREDIMET study results, the control diet was rather high fat diet, as well (37 E%), see the resp. supplemental Table 9 of this PREDIMET publication (New Engl J Med 2018). In fact, the energy % difference was only 4 % (37 vs. 41 %) between the control and intervention diets. Therefore, if the main conclusion on high vs. low fat diets is mainly based on this study, it may not be correct. See also the abstract, lines 205-206 and conclusions, and Table 1 as well! Furthermore, most controlled intervention trials in the prevention of T2D have been carried out with rather low-fat diets.

Response: Thanks for pointing this out. Our statements in lines 197-198 and 205-206 are not only based on the PREDIMED study (N= 7,447) but also on three other dietary intervention trials, including the CARDIOPREV (N= 1,002), Shahbazi et al 2018 (N= 336) and WHI-DM trials (N= 48,835). When we state that “most controlled intervention trials in the prevention of T2D have been carried out with rather low-fat diets”, we refer to the multiple-component lifestyle interventions (i.e., physical activity, dietary, and body weight loss intervention), as the reviewer also alludes to. However, when it comes to dietary interventions for the prevention of T2D, these include high-fat diets.

6. Lines 208-210, If there are concerns with some intervention trials, why then they were included in analyses? I refer to many meta-analyses applying GRADE

on the subject. Regarding lifestyle interventions, a priori they may not be blind in terms of the intervention arms. The authors may give more information in this regard in the text.

Response: Thank you for this relevant consideration. In this paper, however, we do not perform a meta-analysis of eligible studies, rather we assess the overall evidence that sociodemographic, clinical, behavioral, and molecular factors affect intervention to prevent T2D. Furthermore, we also assess the quality of each of the studies using the Diabetes Canada Clinical Practice Scale, which grades the certainty of the evidence based on study design, sample size, follow-up, and reproducible outcome measures. As described above, we could not apply a GRADE threshold as no studies were specifically designed and conducted to investigate the effectiveness of T2D prevention strategies according to individuals' characteristics. We believe our discussion covers all these relevant aspects and makes the point that new studies are needed to implement precision diabetes medicine, especially regarding T2D and prevention.

7. Lines 224- gives the results on the effect of initial BMI on weight loss success but does not give any analysis on the degree of body weight loss on the outcome measure i.e., incident diabetes. In this regard, the conclusions may be different as I pointed out above.

Response: Lines 224-230 elucidate the role of BMI as an effect modifier of lifestyle and dietary interventions in T2D prevention. Analysing the degree of body weight loss on incident T2D was beyond the scope of this review. We only focussed on potential moderators of lifestyle interventions. [See also our response to the Reviewer's main concern earlier.]

8. Regarding the effect of genetic and other molecular factors, the evidence is scares and further controlled trials are needed with appropriate study designs as the authors also pointed out. Lines 263-270, may be removed or shortened because of lacking confirmation?

Response: As suggested, we have now shortened lines 263-270 as given below:

Page 13, line 279: *“Besides genetics, other molecular markers such as plasma branched-chain amino acids and miRNAs have been studied. The evidence that these molecular features modify the efficacy of dietary interventions in the prevention of T2D has only low to very-low certainty. (Table 5, Figure 2)”*

9. Lines 282- Discussion, even if the initial BMI does not play a role in the success, I think that the authors should comment on the importance of weight reduction in the prevention of T2D. The readers may misunderstand the message in the first paragraph in discussion.

Response: We appreciate the reviewers view on this and have commented on this earlier, in our response to the first comment. In brief, the main aim of this review was to study the factors that modify the effect of lifestyle interventions in the prevention of T2D. Weight reduction could be considered as an important mediator of lifestyle interventions in preventing T2D, not an effect modifier. Hence, we have not included studies that focused on the effect of weight reduction in the prevention of T2D. We have clarified this in our manuscript by adding the following lines.

Page 18, line 373: *Moreover, as our scope only included moderators of the intervention efficacy on T2D, which are typically measured prior to or at baseline²⁵, important mediators of the intervention effects on T2D as e.g. weight loss was not addressed and discussed. This will be important to address in future studies to gain a deeper understanding of heterogenous lifestyle interventions responses..*

10. Lines 293-294, as said above, it is weight loss that matters along with healthy diet and lifestyles like physical activity in the prevention of T2D.

Response: Thanks again for this important point about weight loss. We have addressed this in the previous response.

11. Lines 315-317, I do not totally understand the message of this sentence. Do the authors challenge the evidence of major T2D prevention trials or not? I think we cannot compare the studies carried out with high risk people to those with normal glucose tolerance since these two groups are quite different in terms of the absolute and relative risk of T2D.

Response: Our findings support that people at increased T2D risk slightly benefit more from a lifestyle intervention than those at low risk, which aligns with large-scale trials. We agree with the Reviewer that people with/without prediabetes are very different and that these findings could be attributed to risk magnification. Overall, it supports our decision not to conduct a formal meta-analysis. We have modified this sentence accordingly:

Page 15, line 298: *Individuals with prediabetes at baseline benefit slightly more from prevention interventions than those without prediabetes, but the certainty of the evidence was low. This can be explained by relative and absolute risk differences among people with/without prediabetes.*

12. Ref. 52, there are similar data on the DPS study population regarding TCF7L2 polymorphism, Wang J et al. Diabetologia 2007.

Response: We thank the Reviewer for pointing this out to us. The study by Wang et al on DPS participants showed that *TCF7L2* SNPs rs12255372 and rs7903146 are associated with incidence of T2D in the control group but not in the intervention group.

However, there was no significant interaction between the SNPs and intervention groups. We have now included this study in our review under "studies identified by manual searching".

See the changes made in the Study attrition Diagram (Figure 1), Figure 2, Table 1 and Table 5, Results section (Molecular factors).

13. Discussion, lines 319-329. My view is that although based on limited data published, current evidence suggests that lifestyles matter regardless of heterogenous matter of the disease and that insulin resistance is highly modifiable by weight loss and increased physical activity, and furthermore, very limited evidence suggests that weight loss may also help to maintain insulin secretion capacity or even improve it, see e.g., DiRECT study results and those published from DPS (de Mello V et al. Diabetes Care 2013), and Indian Study (Snehalatha C et al. Diabetes Care 2009) and multiple DPP publications related to the topic. Finally, I totally agree with the authors regarding the main conclusions of the manuscript.

Response: Thank you for this relevant consideration. We agree with the Reviewer that clinical studies have consistently demonstrated that T2D and intermediate phenotypes can be reversed by the adoption of a healthy lifestyle and weight loss. We are also aware of this notion that weight loss should be considered a central target for diabetes prevention regardless of the mechanisms influencing disease risk. While many studies have shown the benefits of losing body weight on the risk of developing T2D or diabetes remission, there is still substantial variability in individual response to weight-loss interventions. For example, in the Diet Intervention Examining The Factors Interacting with Treatment Success (DIETFITS) study, it was shown that weight change varied widely within each study group, ranging from a loss of approximately 30 kg to a gain of approximately 10 kg. In clinical trials for weight loss medication, treatment with subcutaneous 2.4 mg semaglutide plus lifestyle intervention resulted in a mean body weight change of ~15% after 68 weeks, but body weight loss ranged from losing ~45% of initial body weight to gaining ~15%. While weight loss is critical in T2D prevention, these findings reinforce the continued effort to identify molecular, environmental and social characteristics underlying the variable response to diabetes prevention interventions.

We have added the following information:

Page 17, line 359: *While many studies support the benefits of body weight loss on the risk of developing T2D regardless of the mechanisms underlying T2D, there is still substantial variability in individual response to weight-loss interventions. For example, the DIETFITS study²⁴, showed that weight change varied widely within each study group, ranging from a loss of approximately 30 kg to a gain of approximately 10 kg. While weight loss is critical in T2D prevention, these findings reinforce the continued*

effort to identify molecular, environmental and social characteristics underlying the variable response to diabetes prevention interventions.

Reviewer #2:

Prevention of type 2 diabetes is a key public health issue. The initiative outlined here to investigate intervention efficacy based on different study and participant characteristics is of crucial importance. The rationale for this manuscript is therefore reasonable. This is a well written manuscript with generally appropriate methodology, analysis and interpretation. It would benefit from more details with regards to study characteristics to allow further interpretation.

Response: We thank Reviewer #2 for the favourable review of our study and for the helpful suggestions to improve the manuscript.

Comments:

1. Line 81 – A comment on the relatively efficacy of diet or physical activity components on prevention of diabetes (both in association and not in association with weight loss) would be useful here.

Response: Thanks for this valuable suggestion. We have added the following information:

Page 5, line 81: *In many cases, diet and physical activity interventions targeted at body weight reduction or preventing weight gain have been demonstrated to delay progression.³⁻⁶ However, T2D remains a major cause of morbidity and mortality globally.⁷*

2. Line 90 – A comment here on the use (or not) on implementation science in aiding translation and scalability in these interventions would be useful here. The search is reasonably old and I would recommend an update be conducted.

Response: We have addressed these comments.

We have now added a comment as given below. We have also added a recent reference.

Page 5, line 90: *However, T2D incidence has only escalated in the decades since, despite the success of early clinical trials. Thus, implementation strategies for diabetes prevention in the real-world setting involving more practical ways of identifying high-risk individuals and precision prevention research may contribute to understanding this gap.⁸*

3. Methods: The term lifestyle for describing the intervention is used frequently, and then often followed by diet/behavioural etc. I would suggest an upfront definition what constitutes a lifestyle intervention – ie diet, PA? Psychological interventions? This is currently unclear, it is done on line 125 to some extent but I would move this statement upfront in the methods and then just refer to lifestyle after this. Similarly for line 163 I would define this in more detail (ie

lifestyle (and what exactly is meant by this, ie multicomponent), diet alone, supplement etc). In table 2 it is also referred to as ‘intensive’ lifestyle – I would define and justify this term (eg is this based on total session number or another factor)? For example, smoking is referred to for the first time on line 193. And confirming there were no studies for exercise in isolation?

Response: Thanks for highlighting this aspect. Lifestyle interventions were defined as interventions ranging from interventions on single behavioral factors, including diet, physical activity, smoking, and body weight loss, to multi-component modification programs focused on a combination of different behavioral components. We have clarified this in the manuscript as given below and also added more details on line 163. In Tables 2-5, we have now used the term Lifestyle intervention, not intensive lifestyle intervention.

Page 7, line 114: *Our search included MEDLINE, Embase, and Cochrane Central Register of Controlled Trials databases for studies reporting on the efficacy of lifestyle or behavioral interventions with T2D incidence, published from 1/1/2000 to 7/15/2021. Lifestyle interventions were defined as interventions ranging from interventions on single behavioral factors including diet, physical activity, smoking, and body weight loss, to multi-component modification programs focused on a combination of different behavioral components.*

Page 9, lines 172: *We collated the literature according to intervention type as lifestyle intervention programs (single or multicomponent), dietary pattern interventions (involving modifications in diet only), or supplement intervention and effect modifier analyzed (e.g., sex, age strata) to synthesize results.*

4. Please define and reference the certainty of evidence system in the methods

Response: The certainty of evidence was determined using the Diabetes Canada 2018 Clinical Practice Scale (Ref 15). Grades were assigned according to the levels of evidence, which were determined based on study design, sample size, follow-up, and reproducible outcome measures. Grade A indicated the highest certainty of evidence, and Grade D indicated the lowest certainty of evidence. We have included the following information in the Methods section of the manuscript.

Page 9, lines 181: *We used the Diabetes Canada Clinical Practice Scale to assess the certainty of the evidence for a given effect modifier for T2D treatment or prevention studies.¹⁵ A level of evidence was assigned following the approach and criteria described in supplementary table 4. For example, higher levels were assigned if the study was a systematic overview or meta-analysis of high-quality RCTs or an appropriately designed RCT with adequate power to answer the question posed by the investigators. Then, each recommendation was assigned a grade from A to D.*

Supplementary table 4

Level	Criteria	Grading
Level 1A	Systematic overview or meta-analysis of high-quality RCTs  . Comprehensive search for evidence . Authors avoided bias in selecting articles for inclusion . Authors assessed each article for validity . Reports clear conclusions that are supported by the data and appropriate analyses OR Appropriately designed RCT with adequate power to answer the question posed by the investigators  . Patients were randomly allocated to treatment groups . Follow up at least 80% complete . Patients and investigators were blinded to the treatment * . Patients were analyzed in the treatment groups to which they were assigned . The sample size was large enough to detect the outcome of interest 	A
Level 1B	Non-randomized clinical trial or cohort study with indisputable results	A
Level 2	RCT or systematic overview that does not meet Level 1 criteria	B
Level 3	Non-randomized clinical trial or cohort study; systematic overview or meta-analysis of level 3 studies	C
Level 4	Other	D

5. Provide a reference for the justification of including studies with n>100

Response: The rationale for excluding studies with less than 100 participants was internally made as we considered these studies would have limited value for subgroup analyses.

6. Line 125 - I would describe in greater detail what type of effect modification data you accepted here. I.e. statistical interactions? Stratified data? Heterogeneity in response? Any of these or others? The language in the results should then reflect the methods used.

Response: Thanks for this valuable point. Since there is a high discrepancy in reporting effect modification in the current literature, we extracted data on different available measurements, including interaction term estimates, interaction term p-value, stratified estimates, and heterogeneity test. We have now added the following information to clarify this point:

Page 8, line 154: We also extracted data on different available measurements for the interaction of the effect modifier with the intervention effect on T2D, including

interaction term estimates, interaction term p-value, stratified estimates, heterogeneity test and noted any text referring to tests performed with “data not shown”.

7. Suggest reformat Table 3 to be consistent with the Table 1 subheadings of lifestyle/diet/supplement and add in study methodology (RCT or not) and if weight loss was a specific aim of the study or not.

Response: As suggested, Tables 1-5 (pages 28-36 of manuscript) have been reformatted and are consistent in the order of sub-headings: lifestyle interventions, dietary pattern interventions and dietary supplement interventions. Details of whether the trial was RCT or non-randomized or cluster randomized has been mentioned in Table 1, Intervention Design column.

8. I would think a meta-analysis may be possible for some of the subgroup analysis in the Lifestyle interventions as the sample size for number of studies is reasonable (at least for age)?

Response: We appreciate this consideration. As described in our manuscript, we did not perform a meta-analysis because of marked differences in the study populations, interventions and comparators. For example, the 19 studies that included age as an effect modifier were from 15 different intervention trials and the stratification of age groups was different in each study (50-59, 60-69, 70-79; <=67, >67; <=70, >70; <50, >=50; 20-44, 45-64, >=65; <51, 51-61, >61; <45, >=45; <=40y, >40; etc). Only the 4 studies from DPP had the same age group stratification (25-44, 45-59, 60-85). This made meta-analysis not feasible even for age.

9. Line 206 – Please add details on what type of high fat diet

Response: We have now added the details “Mediterranean pattern diet with extra-virgin olive oil/ mixed nuts or high fat diet from olive oil”.

Page 12, line 230: *Available evidence also suggests that a high-fat diet (Mediterranean pattern diet with extra-virgin olive oil/ mixed nuts or high fat diet from olive oil), compared to a low-fat diet, reduces the relative risk of T2D.*

10. Line 209 – generally low quality? Or generally low risk of bias? Unclear.

Response: We thank the reviewer for this important distinction. We have reformatted the sentence accordingly:

Page 12, line 231: *Our certainty of evidence assessment determined that the primary study design and approach was generally of low quality, particularly for the RCTs, owing to randomization methods and uniform outcome assessment.*

11. Line 210 – Blinding is very difficult in lifestyle interventions and I would comment on this. Can you also comment more on if this was blinding of objective or subjective outcome measurement?

Response: We agree with the reviewer that intervention blinding is practically impossible in the context of a lifestyle intervention to prevent T2D. While our evidence appraisal instrument considered whether participants and assessors were blinded to study outcomes (supplementary figure), it is important to clarify that this was not the main reason to grade the evidence as low to very-low. The included studies suffer from low statistical power (i.e., studies not designed for stratification purposes), confounding, and multiple testing. Moreover, these studies were randomized to study the main effect and not the modification effect. Hence, from the point of view of this systematic review, these studies were considered as non-randomized.

12. Line 217 – ‘Evidence presented in studies investigating the effect of a lifestyle intervention according to differences in sociodemographic and clinical characteristics did not indicate statistically different effects for age, sex, race/ethnicity, or socioeconomic status.’ This is unclear as some of the studies in Table 2 had effect modification.

Response: As observed by the Reviewer, some of the included studies claimed differences in the effectiveness of a lifestyle intervention based on specific features. The “Yes/ No” for effect modification, provided in Tables 2-5, is based on significant/nonsignificant effect modification, as reported in the respective studies. However, it is important to be cautious with interpreting these results as these studies suffer from relevant methodological considerations. For example, some studies provided differences among subgroups by analyzing all the intervention groups together, instead of investigating treatment differences in each intervention arm. In addition, the appraisal of evidence suggests low to very low certainty of evidence to indicate different effects for age, sex, race/ethnicity, or socioeconomic status in response to lifestyle intervention. We have now re-written the sentence for better clarity.

Page 11, lines 231: *Certainty of evidence to indicate differential effects for sociodemographic and clinical characteristics such as age, sex, race/ethnicity, socioeconomic status or geographic location in response to lifestyle intervention was low.*

13. Line 224 onwards - Additional supplemental data for numeric data for effect modification data is required including numeric results and the methodology for comparing differences between groups.

Response: Thanks for this consideration. We have now provided a data supplement file showing stratified data with relevant statistical results for each effect modifier. We have added the following information.

Page 11, lines 234: *Study specific numeric estimates for the effect modification are provided in the extended data file.*

14. Line 212 – Which concerns for study design?

Response: The primary goal of this systematic review was to gather and synthesize the evidence supporting differential intervention effects based on sociodemographic, clinical, behavioral, and molecular characteristics. Our systematic search showed the lack of randomized trials specifically designed for such purposes, and only secondary analyses of these clinical trials are available. While these studies are randomized for main effects, the randomization block is not conserved for secondary analysis purposes. For instance, the participants in the Diabetes Prevention Program (DPP) were not randomized based on different fasting glucose values, but the main trial provided estimates according to fasting glucose values. Hence, for the purposes of this systematic review, the DPP can be considered a non-randomized study. In addition, these studies suffer from statistical power (i.e., studies not designed for stratification purposes), confounding, and multiple testing. We have clarified this point in the Methods section.

Page 8, line 140: *The majority of studies (N =76 or 94%) included in this review are RCTs to examine the effect on the intervention on T2D incidence. However, as our focus is on the modification of the intervention effect by sociodemographic, clinical, behavioral and molecular factors, none of these trials can be considered randomized for the purpose of this review, as the randomization block is not conserved.*

15. Line 335 – Which two studies?

Response: Many secondary analyses were derived from two clinical interventions, DPP and the DPS. This has now been added to the manuscript.

Page 18, line 366: *Further, many secondary analyses in this systematic review are derived from two clinical interventions viz, the DPP and the DPS.*

16. Did the studies with clinical effect modifiers comment on if these were independent of BMI (eg prediabetes, age etc). With regards to consideration of underlying clinical conditions, this is particularly relevant given the finding of some suggestion of a modifying effect of prediabetes. This should be

commented on further for specific conditions (eg GDM or others such as PCOS, other pregnancy complications etc). I also note the GDM exclusion criteria, please provide the relevant protocol registration details.

Response: Thanks for this consideration. Eight out of 18 studies have reported that prediabetes status modifies the efficacy of T2D prevention strategies (Table 3). In four of these studies (Ref 16, 18, 86, 91 in manuscript), the effect modification was independent of BMI (p-value for HR significant after adjusting for BMI). The other studies have not mentioned whether the observed modifying effect of prediabetes was independent of BMI.

We appreciate the comment on GDM or other specific pregnancy conditions. The rationale for not including these here is that there is another systematic review of GDM prevention within this series of articles.

(<https://www.medrxiv.org/content/10.1101/2023.04.16.23288650v1>)

The protocol of our systematic review was pre-registered on the International Prospective Register of Systematic Reviews (PROSPERO; CRD42021267686).

17. Table 1 would benefit from a column on if weight loss was an explicit aim of the study

Response: The studies included in our review were from 33 unique intervention trials. Eight of these trials had weight loss along with prevention of T2D as the aim. These are DE-PLAN-CAT, DPP, DPS, Japan DPP, Japanese trial by Kosaka et al, 2004, FIN D2D, PreDE and Zensharen Study for Prevention of Lifestyle Diseases. This information for these trials was reported in Table 1 in the Intervention column. These trials have now been marked with an asterisk symbol (*) in Table 1.

Page 33, Table 1 footnote: **Trials which aimed at weight loss and prevention of T2D*

Reviewers' comments:

Reviewer #1 (Remarks to the Author):

My main concern still is high vs. low fat conclusions since based on the results from the PREDIMET it is impossible to conclude whether the lower diabetes incidence was due to the extra virgin olive oil or nuts vs. a higher fat intake since 4 % difference in fat intake (41 E% vs. 37 E%) may not (alone) explain the observed difference in diabetes incidence between intervention and control groups. The authors should define what they mean by low fat dietary intake/patterns. In my mind the present conclusion is not based on valid scientific data, usually low a fat diet means less than 30 energy % from fat. The authors may also consider the quality of fat that matters. There are some earlier studies (Vessby B et al. Diabetologia 2001) and newer meta-analyses on the topic that suggest it is a higher monoene intake (with a lower SAFA intake) that may benefit in terms of glucose metabolism. Thus, the particular effect of extra-virgin olive oil remains open in the PREDIMET-study.

In the Women's Health Initiative (WHI) study (N > 40 000 participants!), there was no difference in T2D incidence between the two study groups, and subgroup analyses suggested even beneficial effect with a low-fat diet, note that the target of low-fat diet was 20 E % in this study. Thus, the fat intake in WHI study was markedly lower than in the PREDIMET Study where low-fat "control" diet included 37 % of fat. In my mind, these two studies are not comparable with each other. The following is direct conclusion from WHI-study: "Trends toward reduced incidence were greater with greater decreases in total fat intake and weight loss". And next: "Weight loss, rather than macronutrient composition, may be the dominant predictor of reduced risk of diabetes" (Tinker LF et al. Arch Intern Med 2008). Regarding the CARDIOPREV Study, the study published in 2016 did not give any results on the incident diabetes or cardiovascular events.

As for ref. 24 dealing with the same issue I found no evidence that a higher fat (low carb) diet was in the long run (12 months) more beneficial than a healthy low-fat diet. See the direct copy from the text in this report: "Consistent with Figures 2A and 2B, dietary adherence contributed significantly to the success of weight loss. HLC-induced weight loss was significantly associated with a decrease in dietary carbohydrates and increase in dietary fat. In contrast, HLF-induced weight loss was significantly associated with decrease in dietary fat and increase in dietary carbohydrates (Figure 2C; Table S2)". Furthermore, later on in the article it is said: "Interestingly, more participants in the HLF (healthy low fat) group than the HLC (healthy low carb) group achieved long-term weight loss (Figure 4F, p = 0.03)". Thus, I understand that this paper does not support the benefits of HLC with a higher intake of fat, either.

I would like to point out the importance of adherence to lifestyle changes since the adherence to lifestyle changes also tells something about the behavioral factors that matter in the success of prevention of T2D.

Lines 236-239: I am not totally happy with the current conclusion regarding the potential benefit of a high fat diet (even if with low evidence), see above my comments regarding this issue. As I said in my previous review most of the controlled T2D intervention trials in people at increased risk of diabetes where main outcome measure has been incident diabetes, have been carried out with rather low-fat dietary intake/patterns, and better adherence to diet has been shown to associate with the success of prevention of T2D, see e.g., Lindström J et al. Diabetologia 2006. Finally, in DPP and DPS studies, fat intake was reduced by 4- 5 E% in the intervention groups whereas an increase in carbohydrate intake was observed among intervention participants of both studies.

The authors do not give any results on the important component of diet, i.e., dietary fiber that has been consistently associated with lower risk of T2D in observational studies and seems to be an important component of diet when it comes to T2D prevention. At least a comment may be needed.

Minor points, line 62 remove “year”; line 131, sms, clarify; lines 226-227, the message could be clarified.

Response to Reviewers

Reviewer #1 (Remarks to the Author):

My main concern still is high vs. low fat conclusions since based on the results from the PREDIMET it is impossible to conclude whether the lower diabetes incidence was due to the extra virgin olive oil or nuts vs. a higher fat intake since 4 % difference in fat intake (41 E% vs. 37 E%) may not (alone) explain the observed difference in diabetes incidence between intervention and control groups. The authors should define what they mean by low fat dietary intake/patterns. In my mind the present conclusion is not based on valid scientific data, usually low a fat diet means less than 30 energy % from fat. The authors may also consider the quality of fat that matters. There are some earlier studies (Vessby B et al. Diabetologia 2001) and newer meta-analyses on the topic that suggest it is a higher monoene intake (with a lower SAFA intake) that may benefit in terms of glucose metabolism. Thus, the particular effect of extra-virgin olive oil remains open in the PREDIMET-study.

In the Women's Health Initiative (WHI) study (N > 40 000 participants!), there was no difference in T2D incidence between the two study groups, and subgroup analyses suggested even beneficial effect with a low-fat diet, note that the target of low-fat diet was 20 E % in this study. Thus, the fat intake in WHI study was markedly lower than in the PREDIMET Study where low-fat "control" diet included 37 % of fat. In my mind, these two studies are not comparable with each other. The following is direct conclusion from WHI-study: "Trends toward reduced incidence were greater with greater decreases in total fat intake and weight loss". And next: "Weight loss, rather than macronutrient composition, may be the dominant predictor of reduced risk of diabetes" (Tinker LF et al. Arch Intern Med 2008). Regarding the CARDIOPREV Study, the study published in 2016 did not give any results on the incident diabetes or cardiovascular events.

As for ref. 24 dealing with the same issue I found no evidence that a higher fat (low carb) diet was in the long run (12 months) more beneficial than a healthy low-fat diet. See the direct copy from the text in this report: "Consistent with Figures 2A and 2B, dietary adherence contributed significantly to the success of weight loss. HLC-induced weight loss was significantly associated with a decrease in dietary carbohydrates and increase in dietary fat. In contrast, HLF-induced weight loss was significantly associated with decrease in dietary fat and increase in dietary carbohydrates (Figure 2C; Table S2)". Furthermore, later on in the article it is said: "Interestingly, more participants in the HLF (healthy low fat) group than the HLC (healthy low carb) group achieved long-term weight loss (Figure 4F, p = 0.03)". Thus, I understand that this paper does not support the benefits of HLC with a higher intake of fat, either. I would like to point out the importance of adherence to lifestyle changes since

the adherence to lifestyle changes also tells something about the behavioral factors that matter in the success of prevention of T2D.

Lines 236-239: I am not totally happy with the current conclusion regarding the potential benefit of a high fat diet (even if with low evidence), see above my comments regarding this issue. As I said in my previous review most of the controlled T2D intervention trials in people at increased risk of diabetes where main outcome measure has been incident diabetes, have been carried out with rather low-fat dietary intake/patterns, and better adherence to diet has been shown to associate with the success of prevention of T2D, see e.g., Lindström J et al. Diabetologia 2006. Finally, in DPP and DPS studies, fat intake was reduced by 4- 5 E% in the intervention groups whereas an increase in carbohydrate intake was observed among intervention participants of both studies.

Response: We again thank the Reviewer for giving his critical input for improving our manuscript. We have provided our responses to the Reviewer's remarks.

We do not contest that a dietary intervention's main effect may depend on the macronutrient composition of the specific diet in question. Rather, the research question we posed related to if any baseline factors measured prior to initiation of the intervention could modify the effect of an intervention (diet, physical activity, or otherwise) on incident T2D and did not concern if a specific type of intervention was superior to another (e.g., high fat vs. low fat). Also, we do not speculate on the reasons for potential effect modification by baseline factor. The reason for this eventual effect modification could be related to the underlying biology, context, environment, or, as the reviewer acknowledges, the difference in adherence to preventive T2D interventions. One or more of these could, independently or in concert, explain a potential modification. Studies to disentangle contributions are warranted. Therefore, we maintain that the conclusion made in the manuscript is appropriate in relation to the research question. We have reformulated the sentence in line 223.

Page 11, line 223: *Available evidence also suggests that a high-fat diet (Mediterranean pattern diet with extra-virgin olive oil/ mixed nuts or high-fat diet from olive oil) reduces the relative risk of T2D when compared to a diet with a lower amount of fat.*

The authors do not give any results on the important component of diet, i.e., dietary fiber that has been consistently associated with lower risk of T2D in observational studies and seems to be an important component of diet when it comes to T2D prevention. At least a comment may be needed.

Response: Thanks for this point. We would like to emphasize one more time that our systematic review was aimed at investigating factors that could modify the effectiveness of lifestyle interventions for the prevention of T2D. Because there were no studies that investigated whether baseline fiber intake interacts with dietary or

lifestyle interventions that fit our inclusion criteria, we have not discussed the benefits of fiber for the prevention of T2D.

Minor points, line 62 remove “year”; line 131, sms, clarify; lines 226-227, the message could be clarified.

Response: We removed the year accordingly.

Line 131: sms refers to “short message service”. For better clarity, we have replaced “education through sms” with “education through text messaging to mobile phone”.

Response: Thanks; amended as suggested.

REVIEWERS' COMMENTS:

Reviewer #1 (Remarks to the Author):

No further comment